# π-CoT: Prolog-Initialized Chain-of-Thought Prompting for Multi-Hop Question-Answering

## Abstract

Chain-of-Thought (CoT) prompting significantly enhances large language models' (LLMs) problem-solving capabilities, but still struggles with complex multi-hop questions, often falling into circular reasoning patterns or deviating from the logical path entirely. This limitation is particularly acute in retrieval-augmented generation (RAG) settings, where obtaining the right context is critical. We introduce **P**rolog-**I**nitialized **C**hain-**o**f-**T**hought (π-CoT), a novel prompting strategy that combines logic programming's structural rigor with language models' flexibility. π-CoT reformulates multi-hop questions into Prolog queries decomposed as single-hop sub-queries. These are resolved sequentially, producing intermediate artifacts, with which we initialize the subsequent CoT reasoning procedure. Extensive experiments demonstrate that π-CoT significantly outperforms standard RAG and in-context CoT on multi-hop question-answering benchmarks.

## 1 Introduction

Chain-of-thought (CoT) reasoning has emerged as a powerful paradigm for enhancing the problem-solving capabilities of large language models, substantially improving performance on arithmetic, commonsense, and symbolic reasoning tasks (Wei et al., 2022; Kojima et al., 2022). By encouraging models to articulate their reasoning process through intermediate steps, CoT enables more systematic and interpretable problem-solving approaches (Zhang et al., 2022; Wang et al., 2022). However, as the complexity of reasoning tasks increases—particularly in multi-hop scenarios where multiple interconnected inferences must be made—CoT systems have been observed to generalize poorly (Dziri et al., 2023) and become trapped in circular reasoning patterns (Lo et al., 2023; Yao et al., 2023).

This limitation becomes especially pronounced in retrieval-augmented generation (RAG) systems, where CoT excels at single-hop questions that require straightforward document retrieval and reasoning, but struggles significantly with multi-hop queries that demand the integration of information across multiple sources and reasoning steps (Asai et al., 2023). The fundamental challenge lies in CoT's inherent trade-off: while its flexibility allows for creative and adaptive reasoning, this same flexibility can lead to unstructured exploration that fails to maintain logical consistency across complex reasoning chains.

Recent work has explored decomposition-based approaches that break multi-hop questions into manageable single-hop questions (Khot et al., 2023; Zhou et al., 2023; Min et al., 2019). However, even with decomposition, critical gaps remain: models struggle to generate high-quality decompositions without supervision (Patel et al., 2022; Wolfson et al., 2020), fail at reliable fact composition across steps (Press et al., 2023), and lose track of intermediate state in long reasoning chains (Yen et al., 2024; Liu et al., 2024). These failures suggest the need for a more principled reasoning framework that can enforce structure while maintaining flexibility.

In contrast to natural-language reasoning pioneered by CoT, the structured reasoning paradigm has been extensively studied for decades in artificial intelligence and logic programming (Kowalski and Smoliar, 1982; Russell et al., 1995; McCarthy et al., 1960; Simon and Newell, 1971). Prolog, a declarative programming language explicitly designed for structured reasoning tasks, exemplifies this approach through its systematic query resolution mechanisms and logical rule-based inference (Robinson, 1965; Kowalski and Smoliar, 1982; Clocksin and Mellish, 2003). While Prolog's rigid

structure ensures logical consistency, it lacks the flexibility to handle ambiguous natural language, cannot easily incorporate unstructured text from documents, and requires precise logical formulations that may not capture the nuanced reasoning needed for real-world questions.

Recognizing that Prolog and CoT possess complementary strengths and weaknesses, we introduce **Prolog-Initialized Chain-of-Thought ($\pi$-CoT)**, a novel prompting strategy that combines the structural rigor of logic programming with the contextual flexibility of natural language reasoning. Our approach begins by algorithmically reformulating complex multi-hop reasoning questions into equivalent Prolog queries, where each query is deliberately decomposed into a sequence of single-hop sub-queries. These sub-queries are then resolved systematically: each is translated into natural language and posed to a RAG (Lewis et al., 2020; Gao et al., 2023) or in-context CoT system, which retrieves relevant documents and generates answers. The resulting answers are translated back into Prolog facts and incorporated into the evolving knowledge base.

The key insight underlying $\pi$-CoT is that by structuring the reasoning process through Prolog's query resolution mechanism, we ensure that the retrieved context remains highly relevant. Rather than allowing the model to freely explore the reasoning space, potentially losing track of relevant information or pursuing irrelevant tangents (Yao et al., 2023; Dziri et al., 2023), our approach maintains a structured trajectory that systematically builds toward the final answer.

At the completion of the Prolog resolution process, we concatenate the original question, all retrieved documents, and the structured Prolog derivation to create a comprehensive context that initializes the final CoT reasoning step. Because most of the heavy-lifting already happens in the initialization process, the final CoT reasoning is far simpler and more successful.

Through extensive experimental evaluation, we demonstrate that $\pi$-CoT is on par or better than traditional RAG and in-context systems on multi-hop question-answering (QA) benchmarks, including HotpotQA, 2WikiMultiHopQA, MuSiQue, and PhantomWiki. Our results suggest that the principled integration of symbolic reasoning structures with neural language models offers a promising direction for developing more reliable and interpretable reasoning systems.

## 2 RELATED WORKS

**Decomposition for multi-hop question-answering.** Breaking down a complex problem into smaller, manageable parts is a common technique in LLM prompting (Zhou et al., 2023; Khot et al., 2023; Wei et al., 2022). For open-domain QA, Press et al. (2023) prompt the model to generate follow-up questions, and Trivedi et al. (2023) take each new sentence in a CoT as input to the retriever. Importantly, the language model decomposes the question in natural-language steps. Recent works have also explored the use of *explicit plans*, usually in the form of Python programs (Surís et al., 2023; Khattab et al., 2022). While we do not provide a direct comparison due to different model sizes and/or retrieval setups, Monte Carlo Tree Search (Tran et al., 2024), test-time scaling (Wang et al., 2025), and reinforcement learning methods (Li et al., 2025; Jin et al., 2025a; Song et al., 2025) are emerging as promising approaches to open-domain QA.

**Improving language model reasoning with Prolog.** Many prior works generate Prolog from natural language to improve arithmetic reasoning or multi-hop question-answering. Wu and Liu (2025); Vakharia et al. (2024); Borazjanizadeh and Piantadosi (2024) translate questions into Prolog, then query a knowledge base. However, they assume the knowledge base has already been populated with facts. Therefore, they do not address retrieval from unstructured documents. Tan et al. (2024); Yang et al. (2024) utilize Prolog as a source of supervised training signals for math and logical reasoning. Weber et al. (2019) propose a weak unification strategy in Prolog based on semantic similarity. However, their approach requires training and uses pre-defined predicates extracted from the training text. The method closest to our work is that of Chen et al. (2019). They train an LSTM "programmer" to generate programs, which are executed using a BERT-based "reader" to produce answers. In this work, we contribute a training-free strategy and demonstrate its effectiveness with recent LLMs like Llama-3.3-70B-Instruct and Deepseek-R1-Distill-Qwen-32B, using both sparse and dense retrievers. The results of Chen et al. (2019) are also limited by the strictness of a pure Prolog execution.

**Fact extraction and summarization.** Extracting knowledge graph triples from unstructured text is a classical problem in NLP, also known as open information extraction (OpenIE) (Angeli et al.,

2015; Pei et al., 2023; Zhou et al., 2022). To enhance conventional retrieval techniques, LightRAG (Guo et al., 2024) and HippoRAG (Jimenez Gutierrez et al., 2024; Gutiérrez et al., 2025) demonstrate the effectiveness of fact extraction and GraphRAG (Edge et al., 2024) and RAPTOR (Sarthi et al., 2024) propose methods for clustering and summarization. Among these methods, HippoRAG 2 from Gutiérrez et al. (2025) performs the best on HotpotQA, 2WikiMultiHopQA, and MuSiQue. We provide a direct comparison of our method to HippoRAG 2 in the results section below.

## 3 PRELIMINARIES

Dating back to the 1970s, Prolog[1] is a powerful way to represent factual knowledge and perform logical inference (Colmerauer and Roussel, 1996; Russell et al., 1995; Sterling and Shapiro, 1994). Solutions in Prolog are verifiable and compositional, making it particularly well-suited for multi-hop question answering where intermediate steps must be chained reliably.

**Representing factual knowledge.** Consider the the following English sentence about *Harry Potter* (Rowling, 1997):

> "Lockhart is a Defense against the Dark Arts teacher at Hogwarts."

This statement can be represented as the following Prolog **fact**:

```
DADA_teacher("Lockhart", "Hogwarts").
```

In Prolog, `DADA_teacher` is a **predicate** and "Lockhart" and "Hogwarts" are **assignments**. A large amount of real-world knowledge can be encoded in this structured, relational form. In fact, as of early 2025, Wikidata contained over 1.65 billion such knowledge triples.[2] Throughout the paper, we refer to a collection of Prolog facts as a **knowledge base**.

**Multi-hop questions as Prolog queries.** Prolog doesn't just store facts—it also allows us to query them. A **Prolog query** comprises one or more predicates with unassigned variables and asks whether any satisfying variable assignment exists. Consider the following example:

$$\text{"Who is the wife of the Defense against the Dark Arts teacher at Hogwarts} \tag{1}$$
$$\text{who is also a werewolf?"}$$

One way of answering this question involves first identifying all the Defense against the Dark Arts teachers at Hogwarts, then filtering for those who are also werewolves, and finally retrieving the wives of the selected people. Each step depends on resolving the previous step(s), and the intermediate space of possible answers can be large. (Note that J.K. Rowling's books mention seven Defense against the Dark Arts teachers at Hogwarts, so a language model would have to consider seven separate reasoning paths to answer the question.) A concise way of expressing question (1) is the following Prolog query:

```
DADA_teacher(X, "Hogwarts"),
werewolf(X),                           (2)
wife(X, Y),
```

with `Y` representing the answer. Translating questions in natural language to structured queries is a classical problem in the community (Zelle and Mooney, 1996; Zettlemoyer and Collins, 2012).

A **solution** to a Prolog query is a set of variable assignments. Assuming we have a pre-populated knowledge base, the solutions are obtained by executing the query. For example, a knowledge base containing the facts `DADA_teacher("Lockhart", "Hogwarts")`, `werewolf("Lupin")`, and `wife("Lupin", "Tonks")` yields the following solution to query (2):

```
X = "Lupin",
Y = "Tonks".
```

---

[1] We specifically use the SWI-Prolog implementation (Wielemaker et al., 2012) of Prolog due to its rich support of aggregation and if-then-else logic, which allows us to resolve questions like "How many Defense against the Dark Arts teachers have been at Hogwarts?".

[2] https://en.wikipedia.org/wiki/Wikidata

## 4 METHOD: $\pi$-COT — PROLOG-INITIALIZED CHAIN-OF-THOUGHT

Prolog and LLMs are both powerful tools for reasoning, but they have complementary strengths and weaknesses. While Prolog provides a precise and verifiable framework for multi-hop reasoning, it assumes access to a *structured* database, which is often difficult to obtain in practice. In contrast, LLMs can adeptly retrieve and extract information from *unstructured* text, but cannot guarantee logical consistency across reasoning steps.

Motivated by these observations, we combine Prolog and LLMs in two ways: a) we initialize the CoT prompt with the execution trace of a Prolog query. This mitigates LLMs' tendency to diverge from successful reasoning paths for CoT prompting.

b) we retrieve relevant information from available documents with the LLM and generate Prolog facts. This provides Prolog an indirect mechanism to interface natural-language documents and create a structured database.

**Prolog-Initialized CoT.** Our resulting workflow ($\pi$-CoT) is illustrated in Fig. 1. Given the question in natural language, we first prompt[3] the LLM to generate a structured **Prolog query**:

$$Q = (q_1, q_2, \ldots, q_T).$$

Here, $q_i$ is a **sub-query**. The query $Q$ can be resolved step-by-step, one sub-query at a time. By design, the **answer** to the original question lies in one of these sub-queries, usually the last one. For example, the question in eq.(1) yields the Prolog query (2) with $T = 3$ and answer $Y$.

$\pi$-CoT resolves the query step-by-step and logs the execution trace along the way. At step $t$, $\pi$-CoT stores the set of all possible solutions, $S_t$, to the partially formed Prolog query $(q_1, \ldots, q_t)$. Each element in $S_t$ is a dictionary of key-value pairs, where the keys are the names of all variables (e.g. X, Y) in the Prolog query and the values are valid assignments (e.g. Lupin).

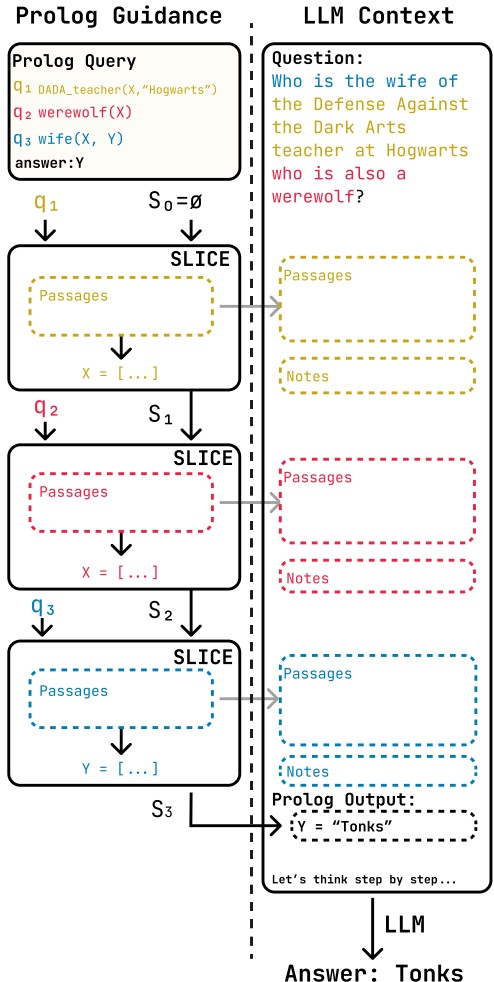

Figure 1: **Overview of $\pi$-CoT.** Left: $\pi$-CoT executes an LLM-generated Prolog query, using the SLICE module to resolve each sub-query $q_t$. Right: $\pi$-CoT uses the passages, notes, and (potentially) answer from the SLICE modules to initialize the CoT prompt for the final LLM call.

### 4.1 SINGLE-STEP EXECUTION WITH SLICE

Each step $t$ follows a fixed procedure, which we refer to as SLICE[4]. Given $S_{t-1}$ from the previous iteration, we resolve the next sub-query $q_t$ and create $S_t$. The sub-query $q_t$ can be of exactly two types: **Verification** (e.g. werewolf(X)), or **Extraction** (e.g. wife(X,Y)). Verification queries reduce the set of solutions in $S_{t-1}$ to those compatible with the query (e.g. removing all teacher names assigned to X that are not werewolves). This scenario is shown in Fig. 2 for sub-query $q_2$. Extraction queries add new variables to $S_t$ that satisfy the query constraint (e.g. adding variable Y and assigning it the names of all teachers' wives already assigned to X). We resolve both query types with the help of the LLM and the available reference documents, while logging the retrieved

---

[3]We provide the query generation prompt in App. B.1.
[4]SLICE stands for Single-step Logical Inference with Contextual Evidence

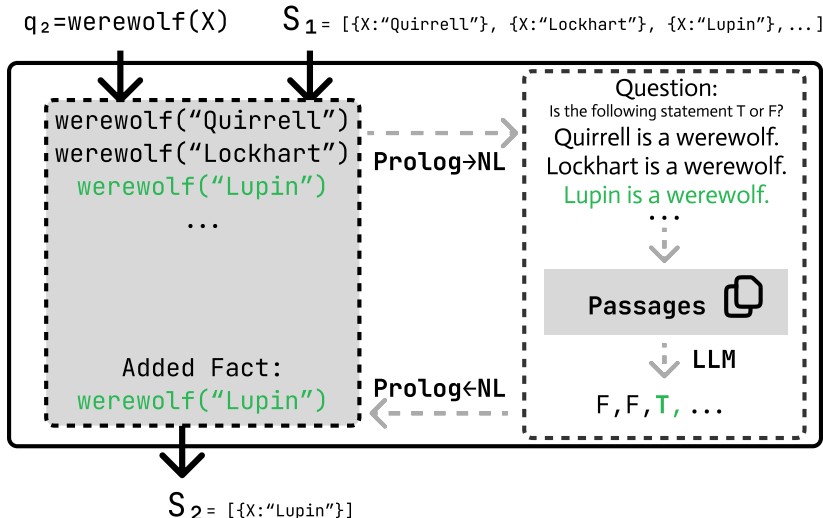

Figure 2: **SLICE module for fact verification in the RAG setting.** At step $t = 2$, the module takes in the previous state $S_1$ containing variable assignments, the current sub-query $q_2$, and the corpus $\mathcal{C}$ (not shown) as inputs and outputs $S_2$. Only the Prolog fact (in green) corresponding to a valid statement is added to a growing knowledge base.

document *passages* and Prolog facts (rephrased in natural language as *Notes*). The Prolog facts are stored in a Prolog knowledge base, allowing us to execute the query $(q_1, \ldots, q_t)$ with an off-the-shelf Prolog interpreter and log its answer. We initialize $S_0 = \emptyset$, as the empty set. We provide additional details of the SLICE module in App. A.

## 4.2 SLICE CHAINING

The inputs to the SLICE module are the sub-query $q_t$, the previous solutions $S_{t-1}$, and the document corpus $\mathcal{C}$. The output of the SLICE module is $S_t$. To derive the final solution, we chain the SLICE modules as follows:

$$S_t = \text{SLICE}(q_t, S_{t-1}, \mathcal{C}) \quad \text{for} \quad t = 1, 2, \ldots, T \quad \text{with} \quad S_0 = \emptyset.$$

By design, the set of solutions is initially empty (i.e., $S_0 = \emptyset$). As the Prolog execution progresses—and more Prolog facts are collected—$S_t$ gets closer to the final solution. After all sub-queries are resolved, $S_T$ is a set of solutions containing the final answer. For example, Fig. 1 shows $S_3 = [\{Y : \texttt{"Tonks"}\}]$ as the final solution.

## 4.3 COMBINING SYMBOLIC AND NATURAL LANGUAGE REASONING

In summary, the iterative process of SLICE chaining generates the following artifacts:

- a collection of retrieved **passages** from each SLICE execution[5];

- a collection of Prolog facts, which we convert into natural language (i.e., "**notes**"); and

- the **answer** from Prolog execution.

The passages contain the necessary factual context to answer the original question. Since these passages may also contain superfluous information, the notes help to focus the model only on relevant information. Due to their iterative construction, these notes also serve to guide the LLM like breadcrumbs toward the final answer. When the final solution is non-empty, the LLM merely needs

---

[5]In the in-context setting, the retrieved passages and original passages are the same.

Table 1: **Accuracy of prompting-based methods on open-domain QA.** We report mean ± 1 standard error for exact match (EM) and F1 score across 500 randomly chosen questions from each dataset. Given a dataset and metric, we perform a repeated measures ANOVA followed by Tukey's HSD with $\alpha = 0.05$ to test significance of paired differences between methods. Bold indicates that no other method performs significantly better.

| Method | HotpotQA | | 2WikiMultiHopQA | | MuSiQue | |
|---|---|---|---|---|---|---|
| | EM ↑ | F1 ↑ | EM ↑ | F1 ↑ | EM ↑ | F1 ↑ |
| Standard RAG | **38.8 ± 2.2** | **52.6 ± 2.0** | 37.2 ± 2.2 | 40.4 ± 2.1 | 11.0 ± 1.4 | 18.1 ± 1.5 |
| Self-Ask | 19.2 ± 1.8 | 28.0 ± 1.8 | 15.8 ± 1.6 | 21.8 ± 1.7 | 5.6 ± 1.0 | 9.1 ± 1.2 |
| IRCoT | **40.4 ± 2.2** | **52.9 ± 2.0** | 32.4 ± 2.1 | 42.5 ± 2.0 | **17.6 ± 1.7** | **24.5 ± 1.8** |
| $\pi$-CoT (Ours) | **42.0 ± 2.2** | **59.1 ± 1.9** | 49.4 ± 2.2 | 57.5 ± 2.1 | 15.2 ± 1.6 | 25.7 ± 1.7 |

Table 2: **Efficiency of prompting-based methods on open-domain QA.** We report mean ± 1 standard error number of BM25 queries, LLM calls, and total tokens for the same questions used in Tab. 1. We further break down the total tokens into prompt and completion tokens in Tab. 7.

| Method | HotpotQA | | | 2WikiMultiHopQA | | | MuSiQue | | |
|---|---|---|---|---|---|---|---|---|---|
| | BM25 | LLM | Tokens | BM25 | LLM | Tokens | BM25 | LLM | Tokens |
| Standard RAG | 1 | 1 | $3.6_{\times 10^3}$ | 1 | 1 | $3.7_{\times 10^3}$ | 1 | 1 | $2.4_{\times 10^3}$ |
| Self-Ask | 3.36 | 3.36 | $1.5_{\times 10^4}$ | 3.44 | 3.44 | $1.6_{\times 10^4}$ | 3.29 | 3.29 | $1.2_{\times 10^4}$ |
| IRCoT | 3.07 | 3.07 | $6.2_{\times 10^4}$ | 3.47 | 3.47 | $5.9_{\times 10^4}$ | 3.51 | 3.51 | $4.5_{\times 10^4}$ |
| $\pi$-CoT (Ours) | 2.82 | 4.82 | $1.8_{\times 10^4}$ | 2.14 | 4.14 | $2.2_{\times 10^4}$ | 3.42 | 5.42 | $1.5_{\times 10^4}$ |

to return it. For the best results, we provide all three artifacts to the LLM and invoke chain-of-thought reasoning to produce the final answer[6].

## 5 MAIN RESULTS

We consider two settings: **(1) open-domain question-answering**, when the corpus is too large to fit within the model's context window, and **(2) in-context question-answering**, when the corpus does fit within the model's context window. The open-domain and in-context QA settings are also known as the *fullwiki* and *distractor* settings in the literature (Yang et al., 2018). To overcome the context limitations in the open-domain QA setting, we use retrieval augmented generation (RAG) to first fetch relevant passages before generation. Since $\pi$-CoT is a prompting method, we require a strong instruction-tuned model and employ the Llama-3.3-70B-Instruct model from Grattafiori et al. (2024). We also provide results utilizing the Deepseek-R1-Distill-Qwen-32B model from (Guo et al., 2025) in the in-context question-answering setting.

### 5.1 OPEN-DOMAIN QUESTION-ANSWERING

We evaluate $\pi$-CoT on three multi-hop QA datasets: (1) HotpotQA from Yang et al. (2018), (2) 2WikiMultiHopQA from Ho et al. (2020), and (3) MuSiQue from Trivedi et al. (2022). Since these datasets are curated from Wikipedia, we can assess Prolog's effectiveness in handling real-world knowledge. For our retrieval setup, we use the preprocessed December 18, 2020 corpus from FlashRAG (Jin et al., 2025b), which contains 20M chunks each of size 100 words, and use BM25 (Robertson et al., 2009) as our retriever. We provide supplementary experiment details in Sec. C.1.

Tab. 1 compares $\pi$-CoT to standard RAG and two multi-hop RAG baselines that also rely on decomposition (via prompting) to handle multi-hop reasoning: **(1) Self-Ask** from Asai et al.

---

[6]See App. B.3 for the prompt template.

Table 3: **Comparison to the state-of-the-art OpenIE method.** We report exact match (EM) and F1 on the splits from Gutiérrez et al. (2025, Tab. 2). We use Llama-3.3-70B-Instruct for generation and NV-Embed-v2 for embedding passages with $k = 5$ per query.

| Method | HotpotQA | | 2WikiMultiHopQA | | MuSiQue | |
| --- | --- | --- | --- | --- | --- | --- |
| | EM ↑ | F1 ↑ | EM ↑ | F1 ↑ | EM ↑ | F1 ↑ |
| Standard RAG | 61.2 | 74.5 | 58.3 | 63.2 | 34.9 | 44.8 |
| HippoRAG 2 (Gutiérrez et al., 2025) | **62.6** | 75.3 | 65.5 | 72.0 | 37.6 | 49.5 |
| $\pi$-CoT (Ours) | 60.3 | **76.8** | **71.1** | **79.6** | **38.5** | **56.2** |

(2023) and (**2**) **IRCoT** from (Trivedi et al., 2023). Among all training-free[7] methods in the FlashRAG repository, IRCoT was the top-performing training-free method on HotpotQA and the second top-performing method on 2WikiMultiHopQA at the time of writing. On HotpotQA, standard RAG, IRCoT, and $\pi$-CoT are comparable in terms of accuracy, with neither method significantly outperforming the other two as determined by a Tukey's HSD test with $\alpha = 0.05$. On 2WikiMultiHopQA, $\pi$-CoT significantly outperforms all other methods in terms of exact match and F1 score. On MuSiQue, IRCoT and $\pi$-CoT achieve comparable accuracy. Despite requiring only a single retriever call, standard RAG achieves surprisingly competitive accuracy on HotpotQA. Min et al. (2019); Chen and Durrett (2019) reveal that multi-hop reasoning is not required for many examples in HotpotQA, possibly explaining our findings. Notably, we find that $\pi$-CoT never performs worse than standard RAG, and does significantly better in the case of 2WikiMultiHopQA and MuSiQue.

To assess the efficiency of the methods in Tab. 1, we measure the number of BM25 queries, the number of LLM calls, and the total token usage. As shown in Tab. 2, $\pi$-CoT uses a similar number of BM25 queries as IRCoT and Self-Ask. We also observe $\pi$-CoT using more LLM calls on average than IRCoT and Self-Ask, which makes sense given that $\pi$-CoT must generate a Prolog query and perform the final chain-of-thought reasoning step. For the price of two extra LLM call, $\pi$-CoT enjoys lower total token usage. In particular, the use of Prolog allows intermediate steps to use separate, short contexts during execution, rather than a single, long context that grows with the number of steps. Our results in Tab. 2 reflect this fundamental difference.

Next, we compare $\pi$-CoT to **HippoRAG 2**, an OpenIE-augmented retrieval method that outperforms GraphRAG (Edge et al., 2024), RAPTOR (Sarthi et al., 2024), and LightRAG (Guo et al., 2024) on multi-hop QA. We follow the experimental setup of Gutiérrez et al. (2025, Tab. 2), which uses the NV-Embed-v2 embedding model of Lee et al. (2024) for retrieval. Instead of using full Wikipedia as the corpus, this experiment uses 9811 passages for HotpotQA, 6119 passages for 2WikiMultiHopQA, and 11656 passages for MuSiQue. Tab. 3 reports mean exact match and F1 score on the provided 1000 samples for each dataset. These results show $\pi$-CoT outperforming both Standard RAG and HippoRAG 2, demonstrating that offline fact extraction is not necessary for good accuracy on multi-hop question-answering tasks.

## 5.2 IN-CONTEXT QUESTION-ANSWERING

When all the necessary information fits in the context window of the model, a natural question is *when does formal reasoning (via $\pi$-CoT) benefit reasoning in natural-language (via CoT)?*. To investigate this, we employ the distractor variants of HotpotQA, 2WikiMultiHopQA, and MuSiQue, where the gold passages are presented in-context alongside a small number of irrelevant ("distractor") passages. We also include two versions of the PhantomWiki benchmark from Gong et al. (2025): PW-S and PW-M. Unlike the other three datasets that are curated from Wikipedia, PhantomWiki generates challenging multi-hop questions from fictional universes to ensure contamination-free LLM evaluation. We provide supplementary experiment details in Sec. C.2.

Tab. 4(a) shows that on HotpotQA, 2WikiMultiHopQA, and MuSiQue, there is no significant difference in accuracy between CoT and $\pi$-CoT. Providing the gold passages to model makes the task considerably easier than considering all of Wikipedia (Min et al., 2019). Thus, Llama-3.3-70B-Instruct

---

[7]Fine-tuned approaches are currently the state-of-the-art on HotpotQA, 2WikiMultiHopQA, and MuSiQue.

Table 4: **Accuracy of prompting-based methods on in-context QA.** We report exact match (EM) and F1 score on HotpotQA, 2WikiMultiHopQA, MuSiQue, PhantomWiki with a corpus size of 50 articles (PW-S), and PhantomWiki with a corpus size of 500 articles (PW-M). Bold indicates that $\pi$-CoT significantly outperforms CoT ($p < 0.05$) according to a paired samples $t$-test given a dataset and metric. We report the computational cost in Tab. 8.

| Method | HotpotQA | | 2WikiMultiHopQA | | MuSiQue | |
|---|---|---|---|---|---|---|
| | EM ↑ | F1 ↑ | EM ↑ | F1 ↑ | EM ↑ | F1 ↑ |
| *Llama-3.3-70B-Instruct* | | | | | | |
| CoT | 62.4 ± 2.2 | 77.4 ± 1.6 | 76.4 ± 1.9 | 83.9 ± 1.5 | 51.2 ± 2.2 | 63.7 ± 1.9 |
| $\pi$-CoT | 58.0 ± 2.2 | 77.7 ± 1.5 | 72.8 ± 2.0 | 82.5 ± 1.5 | 46.4 ± 2.2 | 63.1 ± 1.9 |
| *DeepSeek-R1-Distill-Qwen-32B* | | | | | | |
| CoT | 57.0 ± 2.2 | 74.2 ± 1.7 | 75.2 ± 1.9 | 82.9 ± 1.6 | 45.8 ± 2.2 | 57.3 ± 2.0 |
| $\pi$-CoT | 56.8 ± 2.2 | 75.2 ± 1.6 | 74.6 ± 1.9 | 84.2 ± 1.5 | 46.0 ± 2.2 | 59.6 ± 2.0 |

(a) Real-world (Wikipedia-based) multi-hop QA datasets

| Method | PW-S | | PW-M | |
|---|---|---|---|---|
| | EM ↑ | F1 ↑ | EM ↑ | F1 ↑ |
| *Llama-3.3-70B-Instruct* | | | | |
| CoT | 52.8 ± 1.3 | 71.9 ± 1.0 | 27.5 ± 1.2 | 41.7 ± 1.1 |
| $\pi$-CoT | **78.8 ± 1.1** | **91.4 ± 0.6** | **31.1 ± 1.2** | **56.9 ± 1.0** |
| *DeepSeek-R1-Distill-Qwen-32B* | | | | |
| CoT | 54.4 ± 1.3 | 75.6 ± 0.9 | 17.5 ± 1.0 | 28.0 ± 1.0 |
| $\pi$-CoT (Ours) | **82.7 ± 1.0** | **88.2 ± 0.8** | 18.4 ± 1.0 | **31.1 ± 1.0** |

(b) Synthetic multi-hop QA datasets

with CoT may be hitting a performance ceiling. On PW-S, Llama-3.3-70B-Instruct with CoT achieves an F1 score of 71.9 ± 1.0%. This increases to 91.4 ± 0.6% with $\pi$-CoT. On PW-M, which has a corpus 10 times the size of PW-S, the performance of Llama-3.3-70B-Instruct with CoT drops to 41.7 ± 1.1% F1. We posit this drop is mainly due to the inherent challenges of long-context retrieval. On PW-M, $\pi$-CoT significantly boosts the F1 score to 56.9 ± 1.0 F1—a relative gain of 36%. In Tab. 4(b), we report similar findings using Deepseek-R1-Distill-Qwen-32B as the language model.

Finally, to understand where the gains on PW-S and PW-M come from, we plot accuracy versus the ground-truth difficulty level that comes associated with each question. Gong et al. (2025) defines this difficulty level as the number of hops required to answer the question. According to Fig. 3, the accuracy of $\pi$-CoT and CoT is similar for questions with low difficulty. However, $\pi$-CoT diverges from CoT as the difficulty increases. Since higher difficulty questions require traversing more reasoning paths than lower difficulty questions, our results show that $\pi$-CoT is better able to keep track of multi-hop, multi-branch reasoning than CoT.

## 6 ABLATION ANALYSIS

In this section, we want to analyze how each component in $\pi$-CoT contributes to answering the multi-hop question.

**Contributions of the Prolog component.** Note that when Prolog execution succeeds, the LLM in the final CoT step is instructed to simply copy the Prolog answer as the final output. This setup naturally induces the following categorization: Prolog execution either returns a non-empty solution set ($S_T \neq \emptyset$) or an empty one ($S_T = \emptyset$), and the final CoT answer is either correct ($\pi$-CoT ✓) or

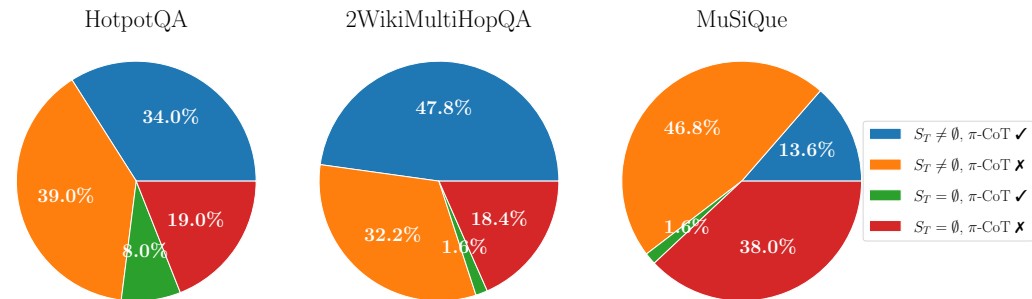

Figure 4: **Analysis of Prolog component based on execution success.** $S_T \neq \emptyset$ means that Prolog outputs an answer, while $S_T = \emptyset$ means that Prolog does not output an answer. $\pi$-CoT ✓ means that $\pi$-CoT generated the ground-truth answer exactly, while $\pi$-CoT ✗ means that $\pi$-CoT generated an incorrect answer. We use the results from Tab. 1.

incorrect ($\pi$-CoT ✗). Here, an answer is considered correct only if it exactly matches the ground-truth answer.

Fig. 4 shows a breakdown of the predictions from Tab. 1 across these four cases. Across datasets, Prolog execution returns an answer in 73% (HotpotQA), 80% (2WikiMultiHopQA), and 60.4% (MuSiQue) of cases. When Prolog returns an answer, $\pi$-CoT is consistently more accurate (46.6%, 60.0%, 22.5%) than the best baseline method in Tab. 1 (40.4%, 37.2%, 17.6%). Even when Prolog sometimes does not return an answer, using the artifacts coming from the Prolog execution can help lead to a correct final answer. Notably, 7.8% of the questions in HotpotQA can be answered correctly this way. For the remaining (27%, 20%, 39.6%) cases, we provide a detailed categorization of the failures in App. E.

**Contributions of components in the final CoT prompt.** We remove specific components from the prompt when generating the final answer (Sec. 4). Tab. 5 shows that removing the passages leads to a (4.7%, 1.2%, 2.4%) drop in F1. This suggests that most of the information needed to answer the question already lies in the notes and Prolog answer. We next remove

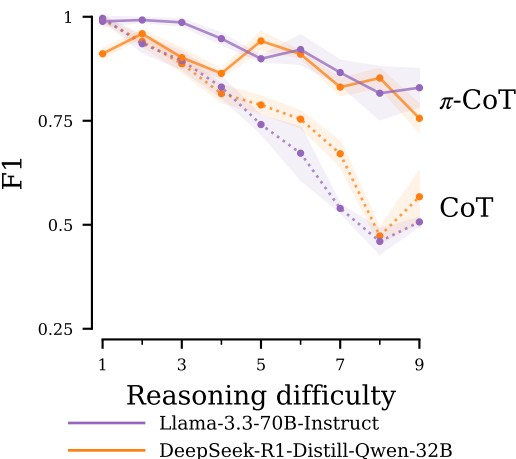

Figure 3: **F1 score vs. difficulty, as measured by number of reasoning steps.** We use the synthetic PW-S benchmark from Gong et al. (2025) and display mean ± 1 standard error. For each model, we evaluate CoT and $\pi$-CoT prompting.

the notes and see performance drop by (4.8%, 0.6%, 0.8%) F1. On HotpotQA especially, notes are important for fuzzy matching when exact string matching fails. Finally, we remove the Prolog answer and see performance drop to near zero for 2WikiMultiHopQA and MuSiQue. This demonstrates the utility of the Prolog answer to the final chain-of-thought reasoning step. Interestingly, Llama-3.3-70B-Instruct is able to achieve 24.1% F1 solely using its internal knowledge. We did not find that including the passages to pose a challenge with long context. Thus, we included all three components for our final model for the best accuracy.

## 7 CONCLUSIONS & FUTURE WORK

In this work, we investigate how formal reasoning can guide reasoning in natural language. We introduce $\pi$-CoT, a novel prompting strategy that initializes the context of an LLM with the intermediate outputs of Prolog-guided execution. Our results show that even strong LLMs like

Table 5: **Ablation analysis of the components in the final chain-of-thought reasoning step.** We use the experimental setup from Tab. 1.

| Method | HotpotQA | | 2WikiMultiHopQA | | MuSiQue | |
|---|---|---|---|---|---|---|
| | EM ↑ | F1 ↑ | EM ↑ | F1 ↑ | EM ↑ | F1 ↑ |
| $\pi$-CoT | 42.0 | 59.1 | 49.4 | 57.5 | 15.2 | 25.7 |
| w/o Passages | 37.2 | 54.4 | 48.4 | 56.3 | 12.6 | 23.3 |
| w/o Notes | 34.0 | 49.6 | 47.8 | 55.7 | 12.4 | 22.5 |
| w/o Prolog Answer | 19.2 | 24.1 | 1.0 | 1.0 | 3.2 | 4.2 |

Llama-3.3-70B-Instruct and Deepseek-R1-Distill-Qwen-32B benefit from being guided through structured reasoning steps, especially on complex, multi-step tasks. More broadly, our work demonstrates the potential of bringing explicit planning and state tracking into language model behavior.

Despite being a purely prompting-based strategy, $\pi$-CoT has potential implications for training future language models to serve as agents that can piece together knowledge across large, dynamic corpora. Inspired by DeepSeek R1 (Guo et al., 2025), significant effort has been made to couple reasoning with retrieval using reinforcement learning (Li et al., 2025; Jin et al., 2025a; Song et al., 2025). An interesting future direction is training language models to generate structured queries instead. $\pi$-CoT provides a way to execute these queries without a pre-existing database. Some steps in $\pi$-CoT may not even require calling an LLM, if the information resides directly in a pre-existing database. Thus, enabling $\pi$-CoT to leverage both unstructured and structured data is a natural next step.

## ETHICS STATEMENT

Our work adheres to the ICLR Code of Ethics, and does not pose any societal, personal, or organizational risks.

## REPRODUCIBILITY STATEMENT

To encourage reproducibility, we use free and open-source software and LLMs. We also include details of our experimental setups in Sec. C. We report sample sizes, standard errors, and random seeds where possible.

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

# A   IMPLEMENTATION DETAILS OF `SLICE` MODULE

**Inputs.** At step $t \in \{1, 2, \ldots, T\}$, `SLICE`[8] takes as input:

- the sub-query, $q_t$;
- the solution from the previous step, $S_{t-1}$; and
- the document corpus, $\mathcal{C}$.

At step $t = 1$, there are no solutions, so $S_0 = \emptyset$. At step $t > 1$, we assign any values in $S_{t-1}$ to the variables in $q_t$. Let's take the second sub-query, $q_2$, from Fig. 1 as an example. As shown in Fig. 2, the previous solution $S_1$ assigns the following values to X: "Quirrell," "Lockhart," "Lupin," etc. These must be substituted into $q_2 = $ `werewolf(X)`, yielding the queries `werewolf("Quirrell")`, `werewolf("Lockhart")`, `werewolf("Lupin")`, etc. to resolve for the current step.

While the query `werewolf("Quirrell")` provides a succinct way of checking whether the claim, "Quirrell was a werewolf," is true, it's still uninterpretable to an LLM. Thus, we need a mechanism to translate Prolog facts to natural-language statements. Our solution is to prompt[9] the LLM to generate a **definition** for $q_t$. Each definition comprises two templates:

- A **question template**, which maps $q_t$ to a question (e.g., "Who are the Defense against the Dark Arts teachers at Hogwarts?").
- A **statement template**, which maps $q_t$ to a claim (e.g., "Lockhart is a Defense against the Dark Arts teacher at Hogwarts.").

Each template serves a different purpose, either *fact extraction* or *fact verification*.

**Fact extraction.** When $q_t$ introduces a new unassigned variable, `SLICE` uses the question template to map $q_t$ to a natural-language question (see **Prolog→NL** in Fig. 2). Next, we call the LLM using chain-of-thought prompting to answer this question (see **LLM** in Fig. 2). We provide the LLM prompt in Sec. B.2. In this work, we consider two strategies to retrieve the relevant evidence for the question:

(S1)  In the **RAG setting**, we use an external retriever to obtain the top-$k$ passages from $\mathcal{C}$.

(S2)  In the **in-context setting**, all passages of $\mathcal{C}$ are provided in the prompt.

Using its innate reading comprehension abilities, the LLM locates the answer to the question from the provided passages and responds with (potentially) multiple answers (e.g., Quirrell, Lockhart, Lupin, etc). These answers are parsed back into Prolog facts (see **Prolog←NL** in Fig. 2) and added to the knowledge base.

**Fact verification.** When all variables in $q_t$ can be assigned values from $S_{t-1}$, we must check that the fact is true. `SLICE` uses the statement template to generate an entailment question (Bowman et al., 2015). For example, `werewolf("Quirrell")` corresponds to the question, "Is the following statement true or false? Quirrell was a werewolf." Equipped with S1 or S2, the LLM answers this question (see **LLM** in Fig. 2). A true claim is added to the knowledge base as a fact; a false claim is simply ignored (and nothing is added to knowledge base). In our running example, the only fact added to the knowledge base at step $t = 2$ is `werewolf("Lupin")` (see green text in Fig. 2).

**Output.** Finally, the output of `SLICE` is the updated solution $S_t$, which can be obtained by querying the knowledge base with $(q_1, \ldots, q_t)$.

# B   PROMPT TEMPLATES

## B.1   PROLOG QUERY AND DEFINITIONS GENERATION

We use the following prompt template for the Prolog query generation of Sec. 4:

---

[8]short for **S**ingle-step **L**ogical **I**nference with **C**ontextual **E**vidence
[9]We provide the definitions generation prompt in Sec. B.1.

```
You will be provided a question. Your goal is to devise a
Prolog query to answer this question. Your response must end in
"**Query:** <query>\n**Target:** <target>\n**Definition:**
<definition>", where <query> is a Prolog query that when
executed, will yield the answer to the question, <target>
is the target variable in the Prolog query to be returned
as the final answer, and <definition> defines the semantic
meaning of predicates in the Prolog query.

Here are some examples:
(START OF EXAMPLES)
{examples}
(END OF EXAMPLES)

Question: {question}
Answer:
```

To form the prompt, `examples` is replaced with few-shot examples specific to each dataset and `question` is replaced with the natural-language question.

## B.2    CHAIN-OF-THOUGHT FACT EXTRACTION AND VERIFICATION

```
You are given the following evidence:
(BEGIN EVIDENCE)
{{evidence}}
(END EVIDENCE)

You will be provided a question. If there is a single answer,
your response must end with the final answer enclosed in tags:
<answer>FINAL_ANSWER</answer>
If there are multiple answers, your response must end with the
final answers enclosed in tags:
<answer>FINAL_ANSWER_1, FINAL_ANSWER_2, ..., FINAL_ANSWER_N</answer>.
If FINAL_ANSWER_N is a string, it must be enclosed in double quotes.
For example, <answer>"FINAL_ANSWER_1", "FINAL_ANSWER_2"</answer>
If FINAL_ANSWER_N is a date, it must be formatted as
date(year, month, day).
If no information is available to answer the question,
your response must end with: <answer></answer>.

Here are some examples:
(START OF EXAMPLES)
{{examples}}
(END OF EXAMPLES)

Question: {{question}}
Answer:
```

To instantiate the prompt, `evidence` is replaced by relevant passages, `examples` is replaced with dataset-specific few-shot examples, and `question` is replaced with the natural-language question (in the case of fact extraction) or entailment question (in the case of fact verification). Both the instructions and few-shot examples encourage the model to format the answer as Prolog literals so that they can be properly inserted into the Prolog knowledge base.

## B.3    $\pi$-COT

```
You are given the following information:
(BEGIN NOTES)
{{notes}}
```

```
(END NOTES)

(BEGIN EVIDENCE)
{{evidence}}
(END EVIDENCE)

You will be provided a question and an answer from a previous
attempt. If the previous answer is not empty
(e.g. <answer>...</answer>), you should copy the answer directly.
If the previous answer is empty (i.e. <answer></answer>),
you should try to answer the question using
the notes and evidence provided. If there is a single answer,
your response must end with the final answer enclosed in tags:
<answer>FINAL_ANSWER</answer>
If there are multiple answers, your response must end with the
final answers enclosed in tags:
<answer>FINAL_ANSWER_1,FINAL_ANSWER_2,...,FINAL_ANSWER_N</answer>.
If no information is available to answer the question,
your response must end with: <answer></answer>.

Here are some examples:
(START OF EXAMPLES)
{{examples}}
(END OF EXAMPLES)
Question: {{question}}
Previous Answer: <answer>{{answer}}</answer>
Answer:
```

## C SUPPLEMENTARY EXPERIMENT DETAILS

### C.1 FULLWIKI EXPERIMENT DETAILS

**Retrieval setup.** We use the `wiki18_100w` corpus from https://huggingface.co/datasets/RUC-NLPIR/FlashRAG_datasets and use the code from https://github.com/RUC-NLPIR/FlashRAG to build our BM25 index. We allow $k = 14$, $k = 16$, and $k = 8$ chunks per retrieval call for HotpotQA, 2WikiMultiHopQA, and MuSiQue, respectively.

**Baseline implementations.** For standard RAG, we use the Python implementation of `CoTRAGAgent` from https://github.com/kilian-group/phantom-wiki and write few-shot examples for each dataset. For Self-Ask, we use the Python implementation and few-shot examples from https://github.com/RUC-NLPIR/FlashRAG. For IRCoT, we use the Python implementation from https://github.com/RUC-NLPIR/FlashRAG and the GPT3 (`code-davincii-002`) few-shot examples from https://github.com/StonyBrookNLP/ircot (see also (Trivedi et al., 2023, App. G)). We set the maximum iterations for Self-Ask and IRCoT to be 4.

**LLM configuration.** We run Llama-3.3-70B-Instruct on 8 A6000s using vLLM (Kwon et al., 2023) and use greedy decoding with maximum generation tokens 4096. We use the full 128K context length.

### C.2 DISTRACTOR EXPERIMENT DETAILS

**LLM configuration.** For the Llama-3.3-70B-Instruct results, we use vLLM running on 8 A6000s and use greedy decoding with maximum generation tokens 4096. For the Deepseek-R1-Distill-Qwen-32B results, we use vLLM running on 6 A6000s and use sampling temperature 0.6, top-p 0.95, and max generation tokens 16384. We use the full 128K context length for both models.

**PhantomWiki dataset.** For the experiment of Tab. 4(b), we use the code at `https://github.com/kilian-group/phantom-wiki` to generate two synethic multi-hop QA datasets. Tab. 6 lists the configurations for PW-S and PW-M. Each dataset has 1500 questions.

Table 6: **Configurations for PhantomWiki dataset generation.**

| Parameter | PW-S | PW-M |
|---|---|---|
| Question depth | 20 | 20 |
| Number of family trees | 10 | 100 |
| Max family tree size | 50 | 50 |
| Max family tree depth | 20 | 20 |
| Mode | Easy | Easy |
| Number of questions per template | 10 | 10 |
| Seeds | {1,2,3} | {1,2,3} |

## C.3 LLM Usage Statement

Large language models were used for proofreading, revising, and literature search. All claims and arguments were drafted and verified by the authors.

## D    Additional Details of Computational Cost

## E    Analysis of Prolog Errors

We manually inspected the results in Tab. 1 and identified the following Prolog errors:

- Prolog Parsing Errors:
  - **Prolog query parsing error:** The LLM-generated Prolog query could not be parsed by our Prolog grammar. This error typically occurs due to missing double quotes (see examples in Tab. 10 and 11).
  - **Extraction or verification parsing error:** the LLM-generated answer to the fact extraction or fact verification step is an invalid Prolog literal. This error typically occurs due to nested double quotes (see Tab. 12).
- Prolog Execution Errors:
  - **Intermediate predicate existence error:** Information is missing to solve at least one of the intermediate sub-queries.
  - **Final predicate existence error:** execution reaches the final step, but no match is returned. For this type, we observe two patterns: 1) genuinely missing information in the final sub-query, 2)

Table 7: **Token cost of open-domain QA.** We report mean ± 1 standard error number of prompt tokens $\mathbf{P}$ (in thousands) and completion tokens $\mathbf{C}$ (in thousands) for the results in Tab. 2. We use the vLLM inference engine with prefix caching enabled and report the number of cached tokens in parentheses next to the prompt tokens.

| | HotpotQA | | 2WikiMultiHopQA | | MuSiQue | |
|---|---|---|---|---|---|---|
| **Method** | $\mathbf{P} \times 10^3$ | $\mathbf{C} \times 10^3$ | $\mathbf{P} \times 10^3$ | $\mathbf{C} \times 10^3$ | $\mathbf{P} \times 10^3$ | $\mathbf{C} \times 10^3$ |
| Standard RAG | 3.46 (0.032) | 0.159 | 3.63 (0.032) | 0.106 | 2.21 (0.073) | 0.136 |
| Self-Ask | 14.5 (10.5) | 0.402 | 15.7 (10.9) | 0.682 | 11.4 (9.06) | 0.750 |
| IRCoT | 62.3 (51.5) | 0.047 | 59.2 (44.6) | 0.053 | 45.3 (38.1) | 0.057 |
| $\pi$-CoT (Ours) | 18.0 (5.07) | 0.483 | 21.1 (4.21) | 0.543 | 14.4 (3.28) | 0.654 |

Table 8: **Computational cost of in-context QA.** We report the mean ± 1 standard error number of prompt tokens **P** (in thousands), completion tokens **C** (in thousands), and LLM calls for the results in Tab. 4. We use the vLLM inference engine and report the number of cached tokens in parentheses next to the prompt tokens when prefix caching was enabled.

| Method | HotpotQA | | | 2WikiMultiHopQA | | | MuSiQue | | |
|---|---|---|---|---|---|---|---|---|---|
| | $P \times 10^3$ | $C \times 10^3$ | Calls | $P \times 10^3$ | $C \times 10^3$ | Calls | $P \times 10^3$ | $C \times 10^3$ | Calls |
| *Llama-3.3-70B-Instruct* | | | | | | | | | |
| CoT | 2.68 | 0.110 | 1 | 2.10 | 0.0845 | 1 | 3.43 | 0.116 | 1 |
| $\pi$-CoT | 12.4 | 0.412 | 4.37 | 11.0 | 0.437 | 4.47 | 16.1 | 0.553 | 4.90 |
| *DeepSeek-R1-Distill-Qwen-32B* | | | | | | | | | |
| CoT | 2.77 | 0.305 | 1 | 2.20 | 0.298 | 1 | 3.57 | 0.520 | 1 |
| $\pi$-CoT | 11.1 | 1.66 | 4.09 | 9.59 | 1.55 | 4.28 | 12.3 | 1.95 | 4.65 |

(a) Real-world (Wikipedia-based) multi-hop QA datasets

| Method | PW-S | | | PW-M | | |
|---|---|---|---|---|---|---|
| | $P \times 10^3$ | $C \times 10^3$ | Calls | $P \times 10^3$ | $C \times 10^3$ | Calls |
| *Llama-3.3-70B-Instruct* | | | | | | |
| CoT | 8.12 (8.09) | 0.400 | 1 | 68.7 (68.6) | 0.375 | 1 |
| $\pi$-CoT (Ours) | 174 (173) | 1.59 | 23.9 | 2800 (2754) | 2.6 | 40 |
| *DeepSeek-R1-Distill-Qwen-32B* | | | | | | |
| CoT | 8.30 (8.26) | 1.42 | 1 | 70.8 (70.6) | 1.13 | 1 |
| $\pi$-CoT (Ours) | 234 (207) | 7.28 | 21.0 | 1260 (1230) | 7.09 | 16.8 |

(b) Synthetic multi-hop QA datasets

| Error Type | HotpotQA | 2WikiMultiHopQA | MuSiQue |
|---|---|---|---|
| Parsing: Prolog query parsing error | 0.8% | 0% | 0.8% |
| Parsing: Execution parsing error | 0.8% | 0% | 0.8% |
| Execution: Intermediate predicate existence error | 16.2% | 19.8% | 37.4% |
| Execution: Final predicate existence error | 9.2% | 0.2% | 0.6% |
| Total errors | 27.0% | 20.0% | 39.6% |

Table 9: **Percentage breakdown of Prolog errors for the results in Tab. 1.** We define the errors in App. E and provide examples for each.

mismatches due to Prolog's strict string equality. The latter can often be resolved when facts are provided back to the model for CoT reasoning. We provide two examples for this type (see examples in Tab. 13 and Tab. 14)

Table 10: **Prolog query parsing error from HotpotQA.** The constant "...Ready for It?" is missing double quotes in the Prolog query.

| Question | ...Ready for It? is a Taylor Swift song from the album scheduled to be released on what date? |
|---|---|
| **Prolog Query** | `album_of_song(...Ready for It?, A1), release_date(A1, A2)` |

Table 11: **Prolog query parsing error from MuSiQue.** Negations (\+) are not allowed by our Prolog parser.

| | |
|---|---|
| **Question** | which professional sports team would you not see play a home game in the arena where the last place Cream performed? |
| **Prolog Query** | `last_performance_venue("Cream", A1),`
`all_professional_sports_teams(A3),`
`\+home_teams(A1, A3)` |

Table 12: **Fact extraction parsing error from HotpotQA.** The double quotes in the generated answer are incorrectly escaped.

| | |
|---|---|
| **Question** | What Cantonese slang term can mean both "ghost man" and to refer to Westerners? |
| **Response** | "Gweilo" or "gwailou" |

Table 13: **Execution error from HotpotQA.** The generated Prolog query involves the sub-query (A1 == A2) and cannot be satisfied by the facts in the knowledge base. Note that the literal "Royal Air Force (RAF)" and the literal "No. 11 Group RAG" are semantically similar, but are not the same under the Prolog operator ==. The ground-truth answer is Royal Air Force.

| | |
|---|---|
| **Question** | What where both Hawker Hurricane and No. 1455 Flight apart of? |
| **Prolog Query** | `part_of("Hawker Hurricane", A1),`
`part_of("No. 1455 Flight", A2), (A1 == A2)` |
| **Facts** | `part_of("Hawker Hurricane",`
`        "Royal Air Force (RAF)")`
`part_of("Hawker Hurricane",`
`        "Royal Yugoslav Air Force (VVKJ)")`
`part_of("Hawker Hurricane",`
`        "Royal Canadian Air Force")`
`part_of("No. 1455 Flight", "No. 11 Group RAF")` |

Table 14: **Execution error from MuSiQue.** The generated Prolog query involves a string unification (A1 = A2) and cannot be satisfied by the facts in the knowledge base. The ground-truth answer is John D. Loudermilk.

| | |
|---|---|
| **Question** | Who wrote turn me on, which was performed by the person who also performed Chasing Pirates? |
| **Prolog Query** | `performer("Chasing Pirates", A1),`
`performer("turn me on", A2), A1 = A2 ->`
`writer("turn me on", A3)` |
| **Facts** | `performer("Chasing Pirates", "Norah Jones")`
`performer("turn me on", "Sean Smith")`
`writer("turn me on", "Greg Lake")`
`writer("turn me on", "Logan Lynn")`
`writer("turn me on", "Joni Mitchell")` |

# F  ADDITIONAL IN-CONTEXT EXPERIMENTS

Table 15: **Accuracy of CoT with majority voting.** We report mean exact match (EM) and F1 score on a subset of the datasets from Tab. 4

| | HotpotQA | | 2WikiMultiHopQA | | MuSiQue | | PW-S | |
|---|---|---|---|---|---|---|---|---|
| # Samples | EM ↑ | F1 ↑ | EM ↑ | F1 ↑ | EM ↑ | F1 ↑ | EM ↑ | F1 ↑ |
| *Llama-3.3-70B-Instruct* | | | | | | | | |
| 1 | 63.0 | 79.8 | 76.8 | 84.2 | 49.6 | 64.3 | 55.73 | 75.43 |
| 2 | 63.2 | 80.3 | 77.0 | 84.1 | 48.4 | 62.7 | 54.00 | 73.95 |
| 4 | 63.4 | 80.5 | 76.0 | 83.4 | 50.2 | 63.9 | 53.93 | 74.39 |
| 8 | 63.2 | 80.2 | 76.0 | 83.3 | 51.0 | 63.8 | 53.13 | 73.83 |
| *DeepSeek-R1-Distill-Qwen-32B* | | | | | | | | |
| 1 | 57.0 | 74.7 | 75.8 | 83.4 | 45.0 | 56.4 | 53.67 | 74.53 |
| 2 | 57.0 | 74.8 | 75.6 | 83.2 | 45.6 | 56.2 | 54.73 | 76.14 |
| 4 | 57.0 | 75.1 | 76.6 | 83.4 | 46.8 | 57.2 | 54.47 | 75.49 |
| 8 | 58.0 | 75.9 | 77.4 | 84.0 | 49.4 | 59.3 | 54.93 | 75.58 |

We compare standard CoT to the simplest inference-time intervention method: CoT with majority voting. We sample 8 evaluations for each question with the hyperparameters in Tab. 16 and compute the performance when majority voting[10] across 2, 4, and 8 of these samples. On HotpotQA, 2WikiMultiHopQA, and MuSiQue, CoT with majority voting over 4 samples is similar in computational cost to $\pi$-CoT (see Tab. 8). Tab. 15 shows that the benefits of repeated sampling over single sampling are marginal to none.

Table 16: Sampling hyperparameters for the results in Tab. 15.

| Model | Temperature | Top-p | Repetition Penalty | Max Output Tokens |
|---|---|---|---|---|
| Llama-3.3-70B-Instruct | 0.6 | 0.9 | 1.0 | 4096 |
| DeepSeek-R1-Distill-Qwen32B | 0.6 | 0.95 | 1.0 | 16384 |

# G  EXAMPLES OF GENERATED PROLOG QUERIES AND DEFINITIONS

We show five examples of generated Prolog queries and definitions on HotpotQA, 2WikiMultiHopQA, MuSiQue, and PhantomWiki. Examples are randomly chosen.

# H  EXAMPLES OF EXECUTION TRACES OF $\pi$-COT

We show the full $\pi$-CoT workflow on one example from each of 2WikiMultiHopQA, and MuSiQue. We use the generated predictions from the experiment of Tab. 1. Examples are randomly chosen.

---

[10]We use the majority definition from https://github.com/EleutherAI/lm-evaluation-harness.

| Question | Query | Target | Definitions |
|---|---|---|---|
| At what age did Cieli di Toscana's singer become blind? | singer("Cieli di Toscana", A1) age_became_blind(A1, A2) | A2 | `singer(<literal1>, <answer>) -> The singer of <literal1> is <answer>. ; Who is the singer of <literal1>?` `age_became_blind(<literal1>, <answer>) -> <literal1> became blind at the age of <answer>. ; At what age did <literal1> become blind?` |
| Cooperative Living Organization is located in a city that is the county seat of what county in Florida? | city("Cooperative Living Organization", A1), county_seat(A1, A2) | A2 | `- city(<literal1>, <answer>) -> The city where <literal1> is located is <answer>. ; What city is <literal1> located in?` `- county_seat(<literal1>, <answer>) -> <literal1> is the county seat of <answer>. ; What county is <literal1> the county seat of?` |
| Repertoire of Pluecido Domingo appeared in an opera written by whom which was based on a play by William Shakespeare? | opera("Repertoire of Pluecido Domingo", A1), writer(A1, A2), based_on(A1, A3) | A2 | `- opera(<literal1>, <answer>) -> The opera in which <literal1> appeared is <answer>. ; What opera did <literal1> appear in?` `- writer(<literal1>, <answer>) -> The writer of the opera <literal1> is <answer>. ; Who wrote the opera <literal1>?` `- based_on(<literal1>, <answer>) -> The opera <literal1> was based on a play by William Shakespeare <answer>. ; What play by William Shakespeare was the opera <literal1> based on?` |
| In what 2 countries did both X-Large and A Bathing Ape have stores? | x_large_countries(A1), bape_countries(A2), intersection(A1, A2, A3) | A3 | `- x_large_countries(<answer>) -> X-Large had stores in <answer> countries. ; In what countries did X-Large have stores?` `- bape_countries(<answer>) -> A Bathing Ape had stores in <answer> countries. ; In what countries did A Bathing Ape have stores?` `- intersection(<literal1>, <literal2>, <answer>) -> The common countries between <literal1> and <literal2> are <answer>. ; What are the common countries between <literal1> and <literal2>?` |
| Edmund Robert Harris is the principal benefactor of a museum located where ? | museum("Edmund Robert Harris", A1), location(A1, A2) | A2 | `- museum(<literal1>, <answer>) -> The museum that <literal1> is the principal benefactor of is <answer>. ; What museum is <literal1> the principal benefactor of?` `- location(<literal1>, <answer>) -> The location of <literal1> is <answer>. ; Where is <literal1> located?` |

Table 17: 5 examples on generated queries and definitions on HotpotQA.

| Question | Query | Target | Definitions |
|---|---|---|---|
| Which film has the director born later, Brutti Di Notte or Bir Türk'E Gönül Verdim? | director("Brutti Di Notte", A1), director("Bir Türk'E Gönül Verdim", A2), date_of_birth(A1, A3), date_of_birth(A2, A4), A3 @> A4 → A5 | A5 | - director(\<literal\>, \<answer\>) -> The director of \<literal\> is \<answer\>. ; Who is the director of \<literal\>? 
 - date_of_birth(\<literal\>, \<answer\>) -> The date of birth of \<literal\> is \<answer\>. ; When was \<literal\> born? |
| Where did M. K. Muthu's father die? | father("M. K. Muthu", A1), place_of_death(A1, A2) | A2 | - father(\<literal\>, \<answer\>) -> The father of \<literal\> is \<answer\>. ; Who is the father of \<literal\>? 
 - place_of_death(\<literal\>, \<answer\>) -> The place of death of \<literal\> is \<answer\>. ; Where did \<literal\> die? |
| What nationality is Julia Parker (Astrologer)'s husband? | husband("Julia Parker (Astrologer)", A1), nationality(A1, A2) | A2 | - husband(\<literal\>, \<answer\>) -> The husband of \<literal\> is \<answer\>. ; Who is the husband of \<literal\>? 
 - nationality(\<literal\>, \<answer\>) -> The nationality of \<literal\> is \<answer\>. ; What is the nationality of \<literal\>? |
| Are the directors of both films *The Snake Brothers* and *Kooky* from the same country? | director("The Snake Brothers", A1) director("Kooky", A2), country_of_citizenship(A1, A3), country_of_citizenship(A2, A4), A3 == A4 → A5 | A5 | - director(\<literal\>, \<answer\>) -> The director of \<literal\> is \<answer\>. ; Who is the director of \<literal\>? 
 - country_of_citizenship(\<literal\>, \<answer\>) -> The country of citizenship of \<literal\> is \<answer\>. ; What is the country of citizenship of \<literal\>? |
| What is the place of birth of the director of film *Kiss The Sky (Film)*? | director("Kiss The Sky (Film)", A1), place_of_birth(A1, A2) | A2 | - director(\<literal\>, \<answer\>) -> The director of \<literal\> is \<answer\>. ; Who is the director of \<literal\>? 
 - place_of_birth(\<literal\>, \<answer\>) -> The place of birth of \<literal\> is \<answer\>. ; Where was \<literal\> born? |

Table 18: 5 examples of generated queries and definitions on 2WikiMultiHopQA.

| Question | Query | Target | Definitions |
|---|---|---|---|
| Who was the mother of the person who found the sacred writings that became the Book of Mormon? | founder_of_book_of _mormon(A1), mother(A1, A2) | A2 | – founder_of_book_of_mormon(<answer>) -> The person who found the sacred writings that became the Book of Mormon is <answer>.  ; Who found the sacred writings that became the Book of Mormon?
– mother(<literal>, <answer>) -> The mother of <literal> is <answer>.  ; Who is the mother of <literal>? |
| Who wrote Turn Me On by the Thinking About You performer? | performer("Thinking About You", A1), writer("Turn Me On", A1, A2) | A2 | – performer(<literal>, <answer>) -> The performer of <literal> is <answer>.  ; Who performed <literal>?
– writer(<literal1>, <literal2>, <answer>) -> The writer of <literal1> by <literal2> is <answer>.  ; Who wrote <literal1> by <literal2>? |
| What did M. King Hubbert's employer announce it was in the process of doing in April 2010? | employer("M. King Hubbert", A1), announcement(A1, "April 2010", A2) | A2 | – employer(<literal>, <answer>) -> The employer of <literal> is <answer>. ; Who is the employer of <literal>?
– announcement(<literal>, <date>, <answer>) -> <literal> announced it was in the process of doing <answer> on <date>.  ; What did <literal> announce it was doing on <date>? |
| What is the experimental satellite being forerunner to communication satellite of INSAT-4CR's manufacturer called? | manufacturer("INSAT-4CR", A1), experimental_satellite (A1, A2) | A2 | – manufacturer(<literal>, <answer>) -> The manufacturer of <literal> is <answer>.  ; Who is the manufacturer of <literal>?
– experimental_satellite(<literal>, <answer>) -> The experimental satellite being the forerunner to the communication satellite of <literal> is <answer>.  ; What is the experimental satellite being the forerunner to the communication satellite of <literal>? |
| The Socialist Autonomous Province of the city where there were mass executions of Danube Swabian population are located in where? | city_with_mass _executions("Danube Swabian", A1), socialist_autonomous _province(A1, A2), location(A2, A3) | A3 | – city_with_mass_executions(<literal>, <answer>) -> The city with mass executions of <literal> population is <answer>.  ; What city had mass executions of the <literal> population?
– socialist_autonomous_province(<literal>, <answer>) -> The Socialist Autonomous Province of <literal> is <answer>. ; What is the Socialist Autonomous Province of <literal>?
– location(<literal>, <answer>) -> <literal> is located in <answer>.  ; Where is <literal> located? |

Table 19: 5 examples on generated queries and definitions on MuSiQue.

| Question | Query | Target | Definitions |
|---|---|---|---|
| How many children does Shelly Reece have? | aggregate_all(count, distinct(child("Shelly Reece", A1)), A2) | A2 | – child(<literal>, <answer>) -> The child of <literal> is <answer>. ; Who is the child of <literal>? |
| How many friends does the friend of the person whose hobby is aerospace have? | hobby(A1, "aerospace"), friend(A1, A2), aggregate_all(count, distinct(friend(A2, A3)), A4) | A4 | – hobby(<answer>, <literal>) -> The hobby of <answer> is <literal>. ; Who is the person whose hobby is <literal>?
– friend(<literal>, <answer>) -> The friend of <literal> is <answer>. ; Who is the friend of <literal>? |
| How many children does the mother of the child of the person whose date of birth is 0985-04-02 have? | dob(A1, "0985-04-02"), child(A1, A2), mother(A2, A3), aggregate_all(count, distinct(child(A3, A4)), A5) | A5 | – dob(<answer>, <literal>) -> The date of birth of <answer> is <literal>. ; Who is the person whose date of birth is <literal>?
– child(<literal>, <answer>) -> The child of <literal> is <answer>. ; Who is the child of <literal>?
– mother(<literal>, <answer>) -> The mother of <literal> is <answer>. ; Who is the mother of <literal>? |
| How many mothers does the friend of the sister of the sister of the child of the parent of the person whose hobby is mineral collecting have? | hobby(A1, "mineral collecting"), parent(A1, A2), child(A2, A3), sister(A3, A4), sister(A4, A5), friend(A5, A6), aggregate_all(count, distinct(mother(A6, A7)), A8) | A8 | – hobby(<answer>, <literal>) -> The hobby of <answer> is <literal>. ; Who is the person whose hobby is <literal>?
– parent(<literal>, <answer>) -> The parent of <literal> is <answer>. ; Who is the parent of <literal>?
– child(<literal>, <answer>) -> The child of <literal> is <answer>. ; Who is the child of <literal>?
– sister(<literal>, <answer>) -> The sister of <literal> is <answer>. ; Who is the sister of <literal>?
– friend(<literal>, <answer>) -> The friend of <literal> is <answer>. ; Who is the friend of <literal>?
– mother(<literal>, <answer>) -> The mother of <literal> is <answer>. ; Who is the mother of <literal>? |
| How many children does the sibling of the brother of the brother of the sister of Jamal Song have? | sister("Jamal Song", A1), brother(A1, A2), brother(A2, A3), sibling(A3, A4), aggregate_all(count, distinct(child(A4, A5)), A6) | A6 | – sister(<literal>, <answer>) -> The sister of <literal> is <answer>. ; Who is the sister of <literal>?
– brother(<literal>, <answer>) -> The brother of <literal> is <answer>. ; Who is the brother of <literal>?
– sibling(<literal>, <answer>) -> The sibling of <literal> is <answer>. ; Who is the sibling of <literal>?
– child(<literal>, <answer>) -> The child of <literal> is <answer>. ; Who is the child of <literal>? |

Table 20: 5 examples on generated queries and definitions on PhantomWiki-S.

**Question**
Did John Updike and Tom Clancy both publish more than 15 bestselling novels?

**Query**
bestselling_novels(John Updike, A1),
bestselling_novels("Tom Clancy", A2),
(A1 @> 15 -> B1 = "Yes" ; B1 = "No"),
(A2 @> 15 -> B2 = "Yes" ; B2 = "No"),
(B1 == "Yes" -> (B2 == "Yes" -> A3 = "Yes" ; A3 = "No") ; A3 = "No")

**Definitions**
bestselling_novels(<literal1>, <answer>) → <literal1> published <answer> bestselling novels. How many bestselling novels did <literal1> publish?

> **Sub-Query 1:** How many bestselling novels did John Updike publish?
> **Retrieved passages:** *John Updike bibliography*, *Tom Wolfe*, *Toward the End of Time*, *John Updike*, *John Updike*
> **Answer:** more than 20
>
> **Sub-Query 2:** How many bestselling novels did Tom Clancy publish?
> **Retrieved passages:** *Duty and Honor*, *Line of Sight (novel)*, *Tom Clancy's EndWar*, *Mike Maden*, *Mark Greaney (novelist)*
> **Answer:** 17

**Notes**
John Updike published more than 20 bestselling novels.
Tom Clancy published 17 bestselling novels.

**Prolog Answer:**

**Final Answer:** Yes

Table 21: One example on the execution trace of a question on HotpotQA.

**Question**
Who is Wisigard's father-in-law?

**Query**
spouse(Wisigard, A1),
father(A1, A2)

**Definitions**
spouse(<literal>, <answer>)' -> The spouse of <literal> is <answer>. ; Who is the spouse of <literal>? father(<literal>, <answer>)' -> The father of <literal> is <answer>. ; Who is the father of <literal>?

> **Sub-Query 1:** Who is the spouse of Wisigard?
> **Retrieved passages:** *Wisigard*, *Wisigard*, *Deuteria*, *Deuteria*, *Theudebert I*
> **Answer:** Theudebert I
>
> **Sub-Query 2:** Who is the father of Theudebert I?
> **Retrieved passages:** *Theudebert of Soissons*, *Theudebert of Soissons*, *Theudebert I*, *Theudebert I*, *Theudebert I*
> **Answer:** Theuderic I

**Notes**
The spouse of Wisigard is Theudebert I.
The father of Theudebert I is Theuderic I.

**Prolog Answer:** Theuderic I

**Final Answer:** Theuderic I

Table 22: One example on the execution trace of a question on 2WikiMultiHopQA.

---

**Question**
In what region of the country containing A Luoi is the birthplace of John Phan located?

---

**Query**
contains("A Luoi", A1),
birthplace("John Phan", A2),
region(A1, A2, A3)

---

**Definitions**
contains(<literal>, <answer>)' -> <literal> is contained in the country <answer>. ;
What country contains <literal>?
birthplace(<literal>, <answer>)' -> The birthplace of <literal> is <answer>. ; What is
the birthplace of <literal>?
region(<literal1>, <literal2>, <answer>)' -> <literal2> is located in the <answer>
region of <literal1>. ; What region of <literal1> is <literal2> located in?

---

**Sub-Query 1:** What country contains A Luoi?
**Retrieved passages:** *A Luoi Camp*, *A Luoi District*, *A Luoi District*, *A Luoi Camp*, *A Luoi District*
**Answer:** Vietnam

**Sub-Query 2:** What is the birthplace of "John Phan"?
**Retrieved passages:** *John Phan*, *Phan Boi Châu*, *Peter C. Phan*, *John Phan*, *Phan Đình Phùng*
**Answer:** Da Nang, Vietnam

**Sub-Query 3:** What region of "Vietnam" is "Da Nang, Vietnam" located in?
**Retrieved passages:** *Da Nang*, *Da Nang Air Base*, *Da Nang University of Economics*, *Hoàng Sa District*, *Da Nang*
**Answer:** South Central Coast

---

**Notes**
A Luoi is contained in the country Vietnam.
The birthplace of John Phan is Da Nang, Vietnam.
Da Nang, Vietnam is located in the South Central Coast region of Vietnam.
**Prolog Answer:** South Central Coast

---

**Final Answer:** South Central Coast

Table 23: 1 example on the execution trace of a question on MuSiQue.

