# OpenReview forum: "$\pi$-CoT: Prolog-Initialized Chain-of-Thought Prompting for Multi-Hop Question-Answering"
_ICLR.cc/2026/Conference — Submitted to ICLR 2026_

### Official Review · Reviewer_42qM · 2025-10-26

**Soundness:** 3
**Presentation:** 3
**Contribution:** 2
**Rating:** 4
**Confidence:** 3

**Summary:**

This paper presents \pi-CoT, a prompting framework that has an LLM first generate Prolog queries from multi-hop natural language questions, then execute each via SLICE to build a KB. SLICE essentially calls LLM for fact extraction or verification. Experiments on multiple benchmarks report some performance improvements over baselines due to the symbolic prolog-guided decomposition and reasoning structures.

**Strengths:**

Overall, the program framing is motivating, and the idea of combining symbolic scaffolding with Prolog queries with LLM for knowledge extraction and reasoning is conceptually straightforward. The intermediate Prolog queries and per-hop passages improve interpretability and reliability over the reasoning trace. Evaluation across multiple datasets establishes the validity of the method. The analysis in Figure 3 particularly indicates the method's stability even as the reasoning step increases.

**Weaknesses:**

1. There already exist many similar approaches generating Prolog from LLMs for improving arithmetic reasoning and multi-hop QA, like "Reliable Reasoning Beyond Natural Language", "ProSLM: A Prolog Synergized Language Model for explainable Domain Specific Knowledge Based Question Answering", "Arithmetic Reasoning with LLM: Prolog Generation & Permutation", "Thought-Like-Pro: Enhancing Reasoning of Large Language Models through Self-Driven Prolog-based Chain-of-Though". A comparison or discussion with these works is missing in the paper.
2. In open-domain results (Table 1, 2) and in-context datasets (Table 4a), \pi-CoT does not significantly outperform the baselines. Why do the gold passages also diminish the value of Prolog's symbolic guidance?
3. No ablation studies to isolate contributions of components like Prolog and SLICE chaining. Consider adding an ablation by removing specific components from the prompt when generating the final answer (Section 4.3).
4. It appears that the predicate definitions, question templates, and statement templates are LLM‑generated with few-shot examples. How often are these definitions incorrect, and how would this affect the downstream performance?

**Questions:**

1. See weakness for details.
2. Issues in writing: Tables 1–2 label the proposed method as “Memento (Ours)” rather than π‑CoT; “Tukey BSD” is likely a typo for Tukey HSD; typo "Quirell" at line 284

---

> ### Author Response · Authors · 2025-11-21
>
> Thank you for the time you’ve spent reviewing our work and for your thoughtful feedback. We address each of your questions below.
>
> $\blacktriangleright$ **Re: Citations**
>
> Thank you for bringing these papers to our attention. We have added them to the Related Works section of the uploaded revision.
>
> $\blacktriangleright$ **Why do the gold passages also diminish the value of Prolog's symbolic guidance?**
>
> Prolog guidance is most helpful when the model must plan multi-hop retrieval and decide what to fetch next. But in HotpotQA, 2WikiMultiHopQA, and MuSiQue (10, 10, and 20 passages, respectively), all evidence is already provided within a short context and easily handled by long-context models, such as Llama-3.3-70B-Instruct and DeepSeek-R1-Distill-Qwen-32B. With the gold passages upfront, planning becomes less important, and the task reduces to extracting facts and combining them. **As shown in Table 15 (see below), CoT with majority voting provides no advantage over single-shot CoT on these in-context datasets, suggesting that further gains likely require fine-tuning.**
>
> | # Samples | HotpotQA EM ↑ | HotpotQA F1 ↑ | 2WikiMultiHopQA EM ↑ | 2WikiMultiHopQA F1 ↑ | MuSiQue EM ↑ | MuSiQue F1 ↑ | PW-S EM ↑ | PW-S F1 ↑ |
> |---|---|---|---|---|---|---|---|---|
> | **Llama-3.3-70B-Instruct** | | | | | | | | |
> | 1 | 63.0 | 79.8 | 76.8 | 84.2 | 49.6 | 64.3 | 55.73 | 75.43 |
> | 2 | 63.2 | 80.3 | 77.0 | 84.1 | 48.4 | 62.7 | 54.00 | 73.95 |
> | 4 | 63.4 | 80.5 | 76.0 | 83.4 | 50.2 | 63.9 | 53.93 | 74.39 |
> | 8 | 63.2 | 80.2 | 76.0 | 83.3 | 51.0 | 63.8 | 53.13 | 73.83 |
> | **DeepSeek-R1-Distill-Qwen-32B** | | | | | | | | |
> | 1 | 57.0 | 74.7 | 75.8 | 83.4 | 45.0 | 56.4 | 53.67 | 74.53 |
> | 2 | 57.0 | 74.8 | 75.6 | 83.2 | 45.6 | 56.2 | 54.73 | 76.14 |
> | 4 | 57.0 | 75.1 | 76.6 | 83.4 | 46.8 | 57.2 | 54.47 | 75.49 |
> | 8 | 58.0 | 75.9 | 77.4 | 84.0 | 49.4 | 59.3 | 54.93 | 75.58 |
>
> $\blacktriangleright$ **Ablation on different components of $\pi$-CoT**
>
> Following your suggestion, we have included an ablation in the uploaded revision (Table 5) to isolate the contributions of the Prolog answer when generating the final answer.
> 1. We first remove the **passages** and observe F1 scores drop (4.7%, 1.2%, 2.4%). This degradation is small compared to the accuracy of $\pi$-CoT, suggesting the passages contain mostly redundant information. (We did not find the passages to pose a challenge with long context understanding, so we included it in the final CoT prompt for the best accuracy.)
> 2. We next remove the **notes** and see F1 drop by (4.8%, 0.6%, 0.8%). For HotpotQA, the notes are important for fuzzy matching when exact string matching fails (see Table 13 in the uploaded revision for an example), so excluding them leads to a notable reduction in accuracy.
> 3. Finally, we remove the **Prolog answer** and observe EM/F1 tanks on 2WikiMultiHopQA and MuSiQue. Interestingly, Llama-3.3-70B-Instruct achieves 24.1% F1 on HotpotQA solely using its internal knowledge.
>
> We hope the added ablation serves as convincing evidence that the Prolog answer is an essential component of the final chain-of-thought reasoning step.
>
> $\blacktriangleright$ **How often are these definitions incorrect?**
>
> In practice, incorrect predicate or template definitions are rare conditional on a valid decomposition. The challenging step is identifying the correct decomposition; once this structure is in place, generating the associated predicate definitions is typically reliable because the LLM can directly reuse spans or paraphrases from the original question. Given the large number of questions, we include 5 randomly chosen examples each from HotpotQA, 2WikiMultiHopQA, and MuSiQue in Tables 17-18 of the uploaded revision to illustrate the quality of the generated queries.
>
> $\blacktriangleright$ **How would this affect the downstream performance?**
>
> With respect to downstream performance, incorrect definitions matter only insofar as they arise from an incorrect decomposition. If the decomposition is wrong, certain SLICE modules will inevitably fail regardless of how the predicates are phrased. Conversely, when the decomposition is correct, the definitions very rarely introduce additional errors. In practice, overall performance is therefore primarily constrained by decomposition accuracy, not by inaccuracies in the predicate or template definitions.
>
> $\blacktriangleright$ **Re: Issues in writing**
>
> Thank you for spotting these typos. We apologize for this oversight!

---

> > ### Comment · Reviewer_42qM · 2025-11-26
> > **Thanks for the response**
> >
> > Dear Authors,
> >
> > Thank you for the explanations and additional experiments. My concerns regarding the ablation and the related work discussion are primarily resolved. Here are a few brief follow-ups for your further clarification:
> >
> > - Following W4, do you have any quantitative results (like in your Table 9) showing the low percentage of errors related to the generation of "predicate definitions, question templates, and statement templates"?
> >
> > - Following Q2 -- "BSD" is not fixed in the revised paper. What does "BSD" mean?

---

> > > ### Author Response · Authors · 2025-11-29
> > >
> > > $\blacktriangleright$ **Follow-up: W4**
> > >
> > > To quantify how often errors arise specifically from incorrect predicate definitions, question templates, or statement templates, we analyzed all **127 / 100 / 190** Prolog execution errors from HotpotQA / 2WikiMultiHopQA / MuSiQue reported in Table 9. Execution can fail for three reasons:
> > > 1. missing passages (i.e., retrieval error),
> > > 2. incorrect decompositions, and
> > > 3. incorrect predicate/template definitions.
> > >
> > > To isolate definition-related failures, we first remove cases attributable to missing passages. We take the questions and their associated Prolog query/definitions from the fullwiki setting and evaluate them in the distractor setting, where the gold passages are provided. Under this setting, failures are more likely due to decomposition or definition errors rather than retrieval. This reduces the error counts to **63 / 23 / 71** for HotpotQA / 2WikiMultiHopQA / MuSiQue.
> > >
> > > Next, we manually annotate cases where the decomposition is valid but the definitions themselves are incorrect. We find **13 / 0 / 18** such errors, corresponding to **2.6% / 0% / 3.6%** of questions from HotpotQA / 2WikiMultiHopQA / MuSiQue.
> > >
> > > Below we provide representative examples of incorrect (LLM-generated) predicate definitions, question templates, and statement templates from HotpotQA. (MuSiQue errors follow similar patterns.)
> > >
> > > ### Example Errors in Predicate Definitions
> > >
> > > _Question:_ “What festival is held every June in Bartlesville, Oklahoma?”
> > >
> > > _Prolog query:_ festival_held_in(“Bartlesville, Oklahoma”, “June”, A1)
> > >
> > > _Definitions:_
> > > ```
> > > - `festival_held_in(<location>, <month>, <answer>)` -> The festival held in <location> every <month> is <answer>. ; What festival is held every <month> in <location>?
> > > ```
> > > Here, the predicate uses argument placeholders other than \<literal\> and \<answer\>. The correct definition should be
> > > ```
> > > - `festival_held_in(<literal1>, <literal2>, <answer>)` -> The festival held in <literal1> every <literal2> is <answer>. ; What festival is held every <literal2> in <literal1>?
> > > ```
> > >
> > > ### Example Errors in Statement/Question Templates
> > >
> > > _Question:_ “Mooreville, Mississippi is located on which Interstate highway that follows US Route 78?”
> > >
> > > _Prolog query:_ interstate_highway_follows(“US Route 78”, A1), passes_through(A1, “Mooreville, Mississippi”)
> > >
> > > _Definitions:_
> > > ```
> > > - `interstate_highway_follows(<literal1>, <answer>)` -> The Interstate highway that follows <literal1> is <answer>. ; What Interstate highway follows <literal1>?
> > > - `passes_through(<literal1>, <literal2>)` -> <literal1> passes through <literal2>. ; What cities or locations does <literal1> pass through?
> > > ```
> > > Here, the question template for the `passes_through(<literal1>, <literal2>)` predicate should be "What Interstate highway passes through \<literal2\>?".
> > >
> > > $\blacktriangleright$ **Follow-up: Q2**
> > >
> > > We apologize again for this oversight. This has been fixed in the uploaded revision.

---

### Official Review · Reviewer_ydFc · 2025-10-29

**Soundness:** 2
**Presentation:** 3
**Contribution:** 3
**Rating:** 4
**Confidence:** 4

**Summary:**

The paper proposes $\pi$-CoT (Prolog-Initialized Chain-of-Thought), a prompting strategy designed to enhance large language models’ (LLMs) reasoning in multi-hop question-answering tasks.

$\pi$-CoT first translates natural language questions into Prolog queries that decompose reasoning into a sequence of sub-queries. Each sub-query is executed through a SLICE module (Single-step Logical Inference with Contextual Evidence), which uses the LLM to extract or verify relevant facts from unstructured text and update a symbolic knowledge base. The intermediate results including facts, retrieved passages, and natural-language notes are then used to initialize the final chain-of-thought reasoning step, combining symbolic rigor with neural flexibility.

Experiments on HotpotQA, 2WikiMultiHopQA, MuSiQue, and PhantomWiki show consistent or improved performance over RAG, IRCoT, and HippoRAG baselines.

**Strengths:**

[S1] Interesting conceptual perspective. The perspective of reliable decomposition with formal method is interesting. It offers a principled way to constrain reasoning trajectories while keep the subtasks manageable for LLMs during multi-hop inference.

[S2] Writing quality is good. The paper is in general well-written and easy to follow. It also appropriately situates itself among prior works (e.g., IRCoT, Self-Ask, GraphRAG, HippoRAG 2).

[S3] Comprehensive evaluation. The evaluation is done on both real-world datasets (HotpotQA, MuSiQue, 2WikiMultiHopQA) and synthetic multi-hop dataset (PhantomWiki), which provides insights from both realistic and controlled scenarios.

**Weaknesses:**

[W1] Improvement can be inconsistent across datasets. On real-world datasets (e.g., HotpotQA, MuSiQue), $\pi$-CoT performs comparably to baselines, with statistical significance only on certain datasets like 2WikiMultiHopQA. The claimed “significant outperforming” does not hold uniformly. Also, I think it would be good to also make clear that only prompting-based methods are compared in the table. As latest SOTA on the datasets are way higher. E.g., finetuned approaches on HotpotQA is ~10% higher. (https://hotpotqa.github.io)

[W2] Potential high cost. The approach expands every query into multiple Prolog sub-queries, and each step involves a separate LLM call plus intermediate reasoning tokens. Although Tab. 2 reports retriever efficiency, the number of total LLM calls and token usage per question is not quantified. This may make $\pi$-CoT expensive. It would be great that analyses can be provided on that front.

[W3] Dependency on accurate semantic parsing. The pipeline critically relies on the LLM’s ability to correctly translate NL questions into Prolog queries and definitions (Sec. A.1). When the LLM fails to produce the right predicates or variable bindings, downstream steps will propagate errors, potentially cancelling out the benefits.

**Questions:**

[Q1] How many total LLM calls or tokens are required per question (vs. standard RAG or IRCoT)? Could parts of the pipeline (e.g., deterministic Prolog execution) be done without invoking the LLM?

[Q2] Are all intermediate steps truly necessary in the LLM prompt for final performance? Have you tried skipping fact-verification or compressing intermediate notes?

[Q3] For queries like “Who is the female Polish scientist who won the Nobel Prize in 1903”, if the query was decomposed to q1= woman(X), q2=scientist(X) etc, then the intermediate states could contain too many entities. How is this type of token explosion handled?

[Q4] I think there might be a typo in the method name in Tables 1 and 2, I think the name Memento was not used elsewhere in the paper.

---

> ### Author Response · Authors · 2025-11-21
>
> Thank you for the time you’ve spent reviewing our work and for your thoughtful feedback. We now address each of your questions and concerns in turn.
>
> $\blacktriangleright$ **W1**
>
> We have revised our claims to more accurately reflect the empirical results. Specifically, we now state that $\pi$-CoT performs **on par with or better** than traditional RAG and in-context prompting systems on multi-hop QA benchmarks. We have also updated Tables 3-4 to explicitly note that all compared systems are **prompting-based approaches**.
>
> $\blacktriangleright$ **How many total LLM calls or tokens are required per question (vs. standard RAG or IRCoT)?**
>
> For the open-domain QA experiment of Table 1, we report the average number of LLM calls and total tokens per question for all methods (see updated version of Table 2 in the revision).
> * For Standard RAG, Self-Ask, and IRCoT, the number of LLM calls is identical to the number of BM25 retrieval steps. For $\pi$-CoT, the total number of LLM calls is simply the number of BM25 queries plus two: one additional call for generating the Prolog query and another for the final CoT step.
> * Importantly, a larger number of calls does **not** imply higher token usage. In fact, across HotpotQA, 2WikiMultiHopQA, and MuSiQue, $\pi$-CoT uses **(70%, 63%, 67%) fewer tokens** than IRCoT. During Prolog guidance, the context to the LLM in SLICE module is isolated from other SLICE models. In contrast, IRCoT uses a single, continually expanding prompt.
>
> $\blacktriangleright$ **Could parts of the pipeline (e.g., deterministic Prolog execution) be done without invoking the LLM?**
>
> Indeed, certain sub-queries can be resolved deterministically without invoking the LLM. The first row of Table 18 shows an example of this from the 2WikiMultiHopQA dataset.
>
> **Question:** Which film has the director born later, Brutti Di Notte or Bir Türk’E Gönül?
>
> **LLM-generated Prolog query:** `director("Brutti Di Notte", A1), director("Bir Türk’E Gönül Verdim", A2), date_of_birth(A1, A3), date_of_birth(A2, A4), A3 @> A4 → A5`
>
> Here the last query `A3 @> A4 → A5` can be done without invoking an LLM as it checks whether the date contained in `A3` comes after the date contained in `A4`.
>
> $\blacktriangleright$ **Q2**
>
> Thank you for the thoughtful question.
> - **Skipping fact verification:** We have not tried this, but we think skipping fact verification will significantly increase the number of spurious solution paths for questions involve multi-branch reasoning, which would hurt performance. In the example "Who is the wife of the Defense against the Dark Arts teacher at Hogwarts who is also a werewolf", skipping fact verification will not filter out the teachers who are not werewolves thus returns the wives of all the DADA teachers.
> - **Compressing intermediate notes:** Given that the token cost of $\pi$-CoT is relatively low (see Table 2) and the notes do not contribute much to this token cost (see Table 21-23 for examples of notes), we included them anyway to minimize information loss. To directly evaluate whether these notes are even needed, we ran an ablation in which we removed the notes entirely from the final CoT prompt. In this variant, the final CoT step receives only the retrieved passages and the Prolog answer. As shown in the table below, performance remains essentially unchanged on HotpotQA and 2WikiMultiHopQA, with a modest drop on MuSiQue. We conduct a more extensive ablation in Table 5 of the revised manuscript.
>
> | Method | HotpotQA |  | 2WikiMultiHopQA |  | MuSiQue |  |
> |:-------|:--------:|:--------:|:--------:|:--------:|:--------:|:--------:|
> |  | **EM** | **F1** | **EM** | **F1** | **EM** | **F1** |
> | $\pi$-CoT | 42.0 ± 2.2 | 59.1 ± 1.9 | 49.4 ± 2.2 | 57.5 ± 2.1 | 15.2 ± 1.6 | 25.7 ± 1.7 |
> | $\pi$-CoT w/o notes | 41.4 ± 2.2 | 58.8 ± 1.9 | 50.2 ± 2.2 | 58.2 ± 2.1 | 12.4 ± 1.5 | 22.4 ± 1.6 |
>
> $\blacktriangleright$ **Q3**
>
> You’re absolutely right that certain decompositions can make execution inefficient. In practice, the token cost seems to be manageable for HotpotQA, 2WikiMultiHopQA, and MuSiQue. Table 2 in the uploaded revision shows that $\pi$-CoT on average uses (18k, 22k, 15k) total tokens per question, well under the total tokens for IRCoT and only modestly more than Self-Ask.
>
> Perhaps closer to the scenario above is the PhantomWiki dataset, which includes questions like “Who is the sibling of the person whose hobby is birdwatching?”. On PW-M with 500 people, there could be multiple people whose hobby is birdwatching. Table 8 shows the token cost and number of LLM calls on PW-M for two long context models—Llama-3.3-70B-Instruct and DeepSeek-R1-Distill-Qwen-32B—is indeed higher for $\pi$-CoT than for CoT. In a future implementation, we could add hard limits to the number of immediate states to guarantee that it doesn’t explode, but potentially at the expense of recall.

---

### Official Review · Reviewer_dNEJ · 2025-11-01

**Soundness:** 1
**Presentation:** 3
**Contribution:** 3
**Rating:** 2
**Confidence:** 4

**Summary:**

This paper introduces π-CoT (Prolog-Initialized Chain-of-Thought), a prompting strategy that integrates Prolog into LLM reasoning. It decomposes complex multi-hop questions into Prolog sub-queries, each solved step-by-step using the SLICE (Single-step Logical Inference with Contextual Evidence) module, and uses the resulting symbolic traces to initialise the final CoT reasoning step. Notably, both the generation of Prolog sub-queries and the SLICE procedure (translation and answering of sub-queries) are performed by LLMs. The authors evaluate their method on the HotpotQA, 2WikiMultiHopQA, MuSiQue, and PhantomWiki datasets.

**Strengths:**

- The paper presents an interesting approach to addressing the limitations of combining CoT and RAG by incorporating neural-symbolic reasoning through the use of Prolog.
- The paper is clearly written, with well-organised explanations and helpful illustrations.

**Weaknesses:**

- The experimental setup appears somewhat arbitrary and selective. For example, in Table 3, the retrieval model differs from that used in Tables 1 and 2. Is there a specific reason for this? It seems that the baselines in Tables 1 and 2 could also be evaluated under the retrieval model of Table 3 for a fairer comparison. Similarly, in Section 5.2 / Table 4, why are the PW-S and PW-M datasets not used in the experiments of Tables 1–3?

- The paper lacks robust analysis regarding the Prolog component. Were the processes of generating, translating, and answering Prolog sub-queries executed without errors? Did the authors observe any parsing or execution errors (as they are done by LLMs) during these steps?

- The empirical evidence for Prolog improving multi-hop reasoning is limited. In Table 1, π-CoT’s advantage over IRCoT on HotpotQA diminishes on MuSiQue. As the authors note (line 374), prior work has shown that many HotpotQA examples can be solved with single-hop reasoning, whereas MuSiQue was explicitly designed to mitigate this issue. Given that, is there sufficient evidence that Prolog-based reasoning truly benefits multi-hop reasoning? Alternatively, could one interpret that while Prolog offers structured neuro-symbolic reasoning, it may introduce constraints that limit performance on multi-hop tasks (for instance, through error propagation across sub-query resolutions)?

**Questions:**

- In Table 2, the authors measure efficiency solely by the number of retriever calls. However, does the proposed method increase the overall computational cost compared to other RAG + CoT baselines, considering the cost of generating Prolog queries and resolving each sub-query through the LLM, as well as the expanded input length and generation overhead? If so, should it perhaps be compared not only to standard CoT but also to inference-time intervention methods with similar computational budgets?

- Why are statements evaluated as false simply ignored? Could incorporating them as additional context in the final CoT reasoning be beneficial?

- (Minor) In Tables 1 and 2, the proposed method is labelled as Memento (Ours)

---

> ### Author Response · Authors · 2025-11-21
>
> Thank you for your detailed and thoughtful feedback. We address each of your questions in turn.
>
> $\blacktriangleright$ **Retrieval models (Table 3 vs Tables 1-2)**
>
> Our goal for Table 3 was to compare against the state-of-the-art GraphRAG-based method, HippoRAG v2. Their original work uses dense retrievers, not the BM25 sparse retriever. For Tables 1-2, we used the setup from FlashRAG, which does not support the recent NV-Embed-v2 retriever model. Within each table all methods share the same retrieval setup.
>
> $\blacktriangleright$ **Why PW-S / PW-M are not used in Tables 1-3**
>
> Tables 1-3 evaluate open-domain QA where the corpus exceeds the LLM context window, so retrieval is necessary. PW-S/M fit entirely within the context windows of Llama-3.3-70B-Instruct and DeepSeek-R1-Distill-Qwen-32B. We could include an experiment where we retrieve passages, but we were worried that it would be regarded as unrealistic. We are happy to include them if you feel it would add value.
>
> $\blacktriangleright$ **Analysis of the Prolog component**
>
> Prolog guidance directly yields an answer in 73%, 80%, and 60.4% of HotpotQA, 2WikiMultiHopQA, and MuSiQue queries. We categorize the remaining cases into parsing and execution errors and find that parsing errors are rare (≤1.6%). Execution errors are more common (e.g., 25.4% on HotpotQA), but 31% of these failures in HotpotQA (see example below) are recoverable by the final CoT step using the passages and notes. We include the full breakdown across these three datasets and error examples in App. E.
>
> $\blacktriangleright$ **Does Prolog actually improve multi-hop reasoning?**
>
> Figure 4 breaks down accuracy based on Prolog execution success.
> 1. When execution succeeds, $\pi$-CoT’s advantage over Standard RAG widens. On the MuSiQue subset with non-empty Prolog answers, $\pi$-CoT achieves 22.5% EM vs. 14.6% (Standard RAG), 8.3% (Self-Ask), 23.5% (IRCoT). This suggests that certain questions that require true multi-hop reasoning benefit from the structure of Prolog-based reasoning.
> 2. When execution fails, the retrieved trace still improves final CoT reasoning (e.g., +1.6% EM on MuSiQue).
>
> MuSiQue remains harder than HotpotQA because generating clean multi-step Prolog queries is challenging; manual inspection (by the authors) of 100 randomly chosen 3–4-hop questions shows that 19 lead to sub-optimal Prolog queries. The benefits of Prolog-based reasoning are most apparent on the synthetic PhantomWiki dataset, where decomposition is easier; consequently, $\pi$-CoT strongly outperforms CoT on PW-S and PW-M. Since $\pi$-CoT currently generates only one decomposition, it may miss better alternatives when multiple decompositions are possible. Running multiple plans in parallel is a potential future direction.
>
> $\blacktriangleright$ **Computational cost**
>
> For the open-domain QA setting of Table 1, we measure the computational cost along the following three axes:
> 1. BM25 queries (Table 2)
> 2. LLM inference calls (Table 2)
> 3. Token cost: total tokens (Table 2), prompt tokens (Table 7), completion tokens (Table 7), cached tokens (Table 7)
>
> $\pi$-CoT makes the fewest BM25 queries among the methods. It requires two extra LLM calls, one to generate the Prolog query and one for the final CoT answer, whereas Self-Ask and IRCoT issue one LLM call per BM25 query. In total token usage, $\pi$-CoT consumes 70%, 63%, and 67% fewer tokens than IRCoT on HotpotQA, 2WikiMultiHopQA, and MuSiQue. $\pi$-CoT’s prompt length is also shorter than IRCoT’s, and its completion length is comparable to Self-Ask.
>
> We report the computational costs of the in-context setting in Table 8. A current limitation of $\pi$-CoT is that it always triggers Prolog query generation and execution, even for questions that vanilla CoT could solve. On harder datasets like PhantomWiki, this extra reasoning helps, but it is unnecessary for simpler cases. A future direction is to improve the accuracy-efficiency trade-off by letting the LLM decide whether to invoke Prolog-based scaffolding.
>
> $\blacktriangleright$ **Comparison to inference-time interventions**
>
> Table 15 compares standard CoT to majority voting (2/4/8 samples). Note that majority voting across 4 samples has a similar token cost as $\pi$-CoT on HotpotQA, 2WikiMultiHopQA, and MuSiQue. Across all datasets, the benefits of repeated sampling over single sampling are marginal to none. We include the details of this experiment in App. F.
>
> $\blacktriangleright$ **Why ignore statements evaluated as false?**
>
> Since the SLICE modules are chained together, spurious intermediate facts can cause Prolog execution to pursue incorrect solution paths. Filtering invalid facts ensures that only solution paths verified by evidence are passed forward, which is especially important for solving questions with many hops such as PhantomWiki (Table 4b). We think including the false statements in the final CoT reasoning step could confuse the LLM, but experiments are needed to validate this hypothesis.

---

### Official Review · Reviewer_iYvj · 2025-11-01

**Soundness:** 3
**Presentation:** 3
**Contribution:** 3
**Rating:** 4
**Confidence:** 3

**Summary:**

This paper improves the multi-hop reasoning abilities of LLMs by using a prolog inspired approach. They propose pi-CoT which first translates the reasoning question into a prolog program and then executes the prolog program in a step-by-step manner using their approach called SLICE which combines LLM-based reasoning with the semantics of prolog execution. They show that pi-CoT outperforms baselines on several datasets due to its ability to better handle multi-hop multi-branch reasoning than baselines.

**Strengths:**

* The paper is clearly written.
* The method is novel and is a useful way to combine the benefits of a symbolic system (prolog) with the knowledge and natural language reasoning abilities of LLMs.
* Results are presented with variances and appear mostly significant.

**Weaknesses:**

* The method’s dependence on prolog potentially limits this method to a specific set of problems.
* It is not clear what Memento refers to in the tables. Bolding in the tables is also confusing. This makes the results very hard to understand.
* Cost/latency is not evaluated.

**Questions:**

* How does the cost of pi-CoT compare to baselines? Does it perform more RAG lookups or more LLM inference calls than the baselines?
* How often were errors due to incorrect formulation of the problem as prolog in the initial step? For problems that are harder to formalize as prolog, this method will inevitably fail.
* How are concepts like negation handled? This seems like it could lead to a blow up of the knowledge base if the prolog program is not written in a smart way.

---

> ### Author Response · Authors · 2025-11-21
>
> Thank you for the time you’ve spent reviewing our work and for your thoughtful feedback. We address each of your questions below.
>
> $\blacktriangleright$ **How does the cost of pi-CoT compare to baselines?**
>
> * **RAG lookups:** In Table 2, we measure the number of BM25 queries in the open-domain QA setting of Table 1. Across all three datasets, $\pi$-CoT issues fewer BM25 queries than IRCoT and also fewer BM25 queries than Self-Ask on HotpotQA and 2WikiMultiHopQA.
>
> * **LLM inference calls:** Table 2 also shows $\pi$-CoT requiring more inference calls on average than the other prompting-based RAG baselines. This makes sense as $\pi$-CoT performs (i) a Prolog-query generation step and (ii) a final CoT answer-generation step. Despite this, **the total token cost of $\pi$-CoT is 70% (HotpotQA), 63% (2WikiMultiHopQA), and 67% (MuSiQue) lower than the next best method, IRCoT**. $\pi$-CoT requires a similar number of LLM inference calls as in the open-domain setting.
>
> We hope our added results in Table 2, 7, and 8 provide a more complete picture of the computational costs of $\pi$-CoT.
>
> $\blacktriangleright$ **How often were errors due to incorrect formulation of the problem as prolog in the initial step?**
>
> Table 9 in the revised manuscripts analyzes the Prolog errors along “parsing errors” and “execution errors”.
> * **Parsing errors**  (i.e., due to syntax or formatting errors) occur relatively infrequently—1.6% (HotpotQA), 0% (2WikiMultiHopQA), 1.6% (MuSiQue) of the time.
> * **Execution errors** arise due to (i) difficulties in decomposing the natural language questions into reasonable sub-queries and/or (ii) retrieving the relevant context in each SLICE module. On HotpotQA, 2WikiMultiHopQA, and PhantomWiki, most questions can be decomposed into high-quality Prolog queries. We show 5 randomly selected examples from each dataset in App. G. After manual inspection (by the authors) of results on MuSiQue, we found that 19 out of 100 randomly chosen 3- or 4-hop questions lead to sub-optimal LLM-generated Prolog queries. As expected, questions with high ambiguity or multiple possible decompositions make it harder for LLMs to generate high-quality decompositions.
>
> $\blacktriangleright$ **How are concepts like negation handled?**
>
> Our current Prolog parser does not allow for negations. An LLM-generated Prolog query with a negation will result in a parsing error (see also the discussion above) and the final LLM call will have no context besides the question. Fortunately, we only observed one instance of the LLM generating a query with a negation (see Table 11).

---

### Author Response · Authors · 2025-11-21

We sincerely thank all the reviewers for their detailed and constructive feedback on our work. We summarize our additions as follows:

## Prolog Analysis
1. **Figure 4: Contributions of the Prolog component to $\pi$-CoT**

We analyze how the Prolog component affects the downstream accuracy of $\pi$-CoT on the open-domain QA experiment of Table 1.

2. **Table 5: Contributions of passages, notes, and Prolog answer in the final CoT prompt to $\pi$-CoT**

We remove one component (passages, notes, Prolog answer) each time in order from the $\pi$-CoT prompt and observe that the performance drops monotonically as we remove more. This suggests that _all_ the components in our final prompt helps to guide the LLM towards the final answer via CoT reasoning.

| Method | HotpotQA |  | 2WikiMultiHopQA |  | MuSiQue |  |
|:-------|:--------:|:--------:|:--------:|:--------:|:--------:|:--------:|
|  | **EM** $\uparrow$ | **F1** $\uparrow$ | **EM** $\uparrow$ | **F1** $\uparrow$ | **EM** $\uparrow$ | **F1** $\uparrow$ |
| $\pi$-CoT | **42.0** | **59.1** | **49.4** | **57.5** | **15.2** | **25.7** |
| $\quad$ w/o Passages | 37.2 | 54.4 | 48.4 | 56.3 | 12.6 | 23.3 |
| $\quad\quad$ w/o Notes | 34.0 | 49.6 | 47.8 | 55.7 | 12.4 | 22.5 |
| $\quad\quad\quad$ w/o Prolog Answer | 19.2 | 24.1 | 1.0 | 1.0 | 3.2 | 4.2 |

3. **Table 9: Detailed analysis of Prolog error modes**

| **Error Type** | **HotpotQA** | **2WikiMultiHopQA** | **MuSiQue** |
|:-------|:--------:|:--------:|:--------:|
| Parsing: Prolog query parsing error | 0.8% | 0% | 0.8% |
| Parsing: Execution parsing error | 0.8% | 0% | 0.8% |
| Execution: Intermediate predicate existence error | 16.2% | 19.8% | 37.4% |
| Execution: Final predicate existence error | 9.2% | 0.2% | 0.6% |
| **Total errors** | **27.0%** | **20.0%** | **39.6%** |

## Computational Cost

1. **Table 2 (updated): BM25 queries, LLM calls, and total tokens in open-domain QA**
2. **Table 7: Detailed breakdown of total token cost (prompt and completion tokens) on open-domain QA**

We report mean ± 1 standard error number of prompt tokens (in thousands) and completion tokens (in thousands) for the results. We use the vLLM inference engine with prefix caching enabled and report the number of cached tokens in parentheses next to the prompt tokens.

| Method | HotpotQA |  | 2WikiMultiHopQA |  | MuSiQue |  |
|:-------|:--------:|:--------:|:--------:|:--------:|:--------:|:--------:|
|  | **Prompt** | **Completion** | **Prompt** | **Completion** | **Prompt** | **C** |
| Standard RAG | 3.46 (0.032) | 0.159 | 3.63 (0.032) | 0.106 | 2.21 (0.073) | 0.136 |
| Self-Ask | 14.5 (10.5) | 0.402 | 15.7 (10.9) | 0.682 | 11.4 (9.06) | 0.750 |
| IRCoT | 62.3 (51.5) | 0.047 | 59.2 (44.6) | 0.053 | 45.3 (38.1) | 0.057 |
| $\pi$-CoT (Ours) | 18.0 (5.07) | 0.483 | 21.1 (4.21) | 0.543 | 14.4 (3.28) | 0.654 |

3. **Table 8: Computational cost of in-context QA.**

We report mean ± 1 standard error number of prompt tokens (in thousands), completion tokens (in thousands), and LLM calls for the results. We use the vLLM inference engine and report the number of cached tokens in parentheses next to the prompt tokens when prefix caching was enabled.

(a) Real-world (Wikipedia-based) multi-hop QA datasets

| Method | HotpotQA |  |  | 2WikiMultiHopQA |  |  | MuSiQue |  |  |
|:-------|:--------:|:--------:|:--------:|:--------:|:--------:|:--------:|:--------:|:--------:|:--------:|
|  | **Prompt** | **Completion** | **Calls** | **Prompt** | **Completion** | **Calls** | **Prompt** | **Completion** | **Calls** |
| _Llama-3.3-70B-Instruct_ | | | | | | | | | |
| CoT | 2.68 | 0.110 | 1 | 2.10 | 0.0845 | 1 | 3.43 | 0.116 | 1 |
| $\pi$-CoT (Ours) | 12.4 | 0.412 | 4.37 | 11.0 | 0.437 | 4.47 | 16.1 | 0.553 | 4.90 |
| _DeepSeek-R1-Distill-Qwen-32B_ | | | | | | | | | |
| CoT | 2.77 | 0.305 | 1 | 2.20 | 0.298 | 1 | 3.57 | 0.520 | 1 |
| $\pi$-CoT (Ours) | 11.1 | 1.66 | 4.09 | 9.59 | 1.55 | 4.28 | 12.3 | 1.95 | 4.65 |

(b) Synthetic multi-hop QA datasets

| Method | PW-S |  |  | PW-M |  |  |
|:-------|:--------:|:--------:|:--------:|:--------:|:--------:|:--------:|
|  | **Prompt** | **Completion** | **Calls** | **Prompt** | **Completion** | **Calls** |
| _Llama-3.3-70B-Instruct_ | | | | | | |
| CoT | 8.12 (8.09) | 0.400 | 1.00 | 68.7 (68.6) | 0.375 | 1 |
| $\pi$-CoT (Ours) | 174 (173) | 1.59 | 23.9 | 2800 (2754) | 2.6 | 40 |
| _DeepSeek-R1-Distill-Qwen-32B_ | | | | | | |
| CoT | 8.30 (8.26) | 1.42 | 1 | 70.8 (70.6) | 1.13 | 1 |
| $\pi$-CoT (Ours) | 234 (207) | 7.28 | 21.0 | 1260 (1230) | 7.09 | 16.8 |

## Improved Presentation

1. We apologize for the typo "Memento" in our previous tables. It should be replaced with $\pi$-CoT.
2. Appendix G: We include 5 randomly chosen examples of Prolog query and definitions generation from HotpotQA, 2WikiMultiHopQA, MuSiQue, and PhantomWiki.
3. Appendix H: We include 1 example from HotpotQA, 2WikiMultiHopQA, MuSiQue to illustrate the complete $\pi$-CoT workflow.

---

### Meta-Review · Area_Chair_eBus · 2026-01-04

**Summary:**

While some reviewers acknowledged a conceptually appealing neuro-symbolic approach that combines Prolog-style decomposition with LLM reasoning for multi-hop QA, some critical concerns are raised: (i) whether Prolog truly improves hard multi-hop reasoning beyond existing baselines, (ii) the robustness and error characteristics of the LLM-generated Prolog component, (iii) computational cost and fairness of comparisons, and (iv) presentation issues and missing related work.

**Reviewer Concerns:**

Some of the reviewers' concerns are addressed by the additional experiments and explanations provided by the authors. For example, the authors added experiments on computational cost, Prolog error analysis, ablation, and component contributions. Some presentation issues have also resolved.

However, there are still major concerns unresolved:
- Strength of empirical evidence for real-world multi-hop gains: Some reviewers (notably dNEJ) may remain unconvinced that Prolog consistently benefits challenging real-world datasets, where gains are modest, and decomposition errors remain non-trivial.
- Method generality and brittleness: Dependence on accurate decomposition and the lack of support for negation remain acknowledged limitations rather than resolved issues.
- Experimental design choices: While justified, the use of different retrieval setups and exclusion of PW-S/M from some tables may still feel ad hoc to the most critical reviewer.

**Reviewer Scores:**

While some reviewers' concerns were partially resolved, e.g., Reviewer 42qM's concerns are partly resolved, some major concerns remain as explained above. For example, I do not think Reviewer dNEJ would change the score significantly even if the reviewer had the chance to participate in the discussion.

---

### Decision · Program_Chairs · 2026-01-26

Reject